# Ethylene promotes SMAX1 accumulation to inhibit arbuscular mycorrhiza symbiosis

Debatosh Das [1,6,8], Kartikye Varshney[2,3,8], Satoshi Ogawa [4,5,7], Salar Torabi[1,2], Regine Hüttl[2], David C. Nelson [4] & Caroline Gutjahr [1,2,3] ✉

Most land plants engage in arbuscular mycorrhiza (AM) symbiosis with *Glomeromycotina* fungi for better access to mineral nutrients. The plant hormone ethylene suppresses AM development, but a molecular explanation for this phenomenon is lacking. Here we show that ethylene inhibits the expression of many genes required for AM formation in *Lotus japonicus*. These genes include strigolactone biosynthesis genes, which are needed for fungal activation, and Common Symbiosis genes, which are required for fungal entry into the root. Application of strigolactone analogs and ectopic expression of the Common Symbiosis gene *Calcium Calmodulin-dependent Kinase* (*CCaMK*) counteracts the effect of ethylene. Therefore, ethylene likely inhibits AM development by suppressing expression of these genes rather than by inducing defense responses. These same genes are regulated by SUPPRESSOR OF MAX2 1 (SMAX1), a transcriptional repressor that is proteolyzed during karrikin signaling. *SMAX1* is required for suppression of AM by ethylene, and SMAX1 abundance in nuclei increases after ethylene application. We conclude that ethylene suppresses AM by promoting accumulation of SMAX1. SMAX1 emerges as a signaling hub that integrates karrikin and ethylene signaling, thereby orchestrating development of a major plant symbiosis with a plant's physiological state.

Roots of ~80% of land plants associate with beneficial soil fungi of the *Glomeromycotina* in a symbiotic association called arbuscular mycorrhiza (AM), which evolved more than 450 million years ago (reviewed in ref. 1). The obligate biotrophic fungi increase plant mineral nutrition, especially with poorly plant-available phosphate, in exchange for photosynthetically-fixed carbon in the form of sugars and lipids (reviewed in ref. 2). Root colonization by AM fungi (AMF) is initiated by a bidirectional molecular dialog between roots and AMF. In phosphate- and/or nitrogen-limited conditions, roots release signaling compounds called strigolactones that induce fungal spore germination and hyphal branching, thereby priming AMF for root colonization[3,4]. In response, AMF release chito-oligosaccharides and lipochito-oligosaccharides (cumulatively called Myc factors) that bind to and

activate plant Lysine Motive (LysM) receptor-like kinases. Together with SYMBIOSIS RECEPTOR KINASE (SYMRK), a malectin-like leucine-rich repeat receptor kinase, the LysM proteins trigger a signaling cascade, called Common Symbiosis signaling, that reprograms plant cells for symbiosis (summarized in ref. 5). Hallmarks of this signaling cascade are nuclear calcium oscillations, which require two potassium channels (CASTOR and POLLUX), calcium channels of the CNGC15 family[6], and three nuclear porins (NENA, NUP85, and NUP133) (reviewed in ref. 7). The calcium oscillations are thought to be interpreted by a nuclear-localized CALCIUM and CALMODULIN-DEPENDENT KINASE (CCaMK). CCaMK phosphorylates CYCLOPS, a central transcriptional regulator, initiating transcriptional changes that are required for the accommodation of fungal structures inside

[1]Faculty of Biology, Genetics, LMU Munich, Grosshaderner Str. 2-4, Martinsried, Germany. [2]Plant Genetics, TUM School of Life Sciences, Technical University of Munich (TUM), Emil Ramann Str. 4, Freising, Germany. [3]Max-Planck-Institute of Molecular Plant Physiology, Am Mühlenberg 1, Potsdam-Golm, Germany. [4]Department of Botany & Plant Sciences, University of California, 900 University Avenue, Riverside, CA, USA. [5]RIKEN Center for Sustainable Resource Science, Yokohama, Japan. [6]Present address: Redox Bio-Nutrients, 130 S 100 W, Burley, Idaho, USA. [7]Present address: Institute for Chemical Research, Kyoto University, Gokasho, Uji, Kyoto, Japan. [8]These authors contributed equally: Debatosh Das, Kartikye Varshney. ✉e-mail: gutjahr@mpimp-golm.mpg.de

root cells[8]. CYCLOPS interacts with DELLA protein(s), making AM symbiosis vulnerable to signals that alter DELLA abundance such as the hormone gibberellin[9]. *SYMRK*, *CCaMK*, *CYCLOPS*, and *DELLA* are termed 'Common Symbiosis (Sym) Genes', as mutations in these genes abolish root colonization by AMF as well as by nitrogen-fixing rhizobia in legumes[10].

The expression of strigolactone biosynthesis genes and some of the Common Symbiosis genes is increased during phosphate starvation by MYB transcription factors called PHOSPHATE STARVATION RESPONSE[11]. Genetic evidence suggests that the expression of these genes is in turn repressed by SUPPRESSOR OF MAX2 1 (SMAX1), a proteolytic target of karrikin (KAR) signaling[12]. KARs are a class of butenolide compounds found in smoke of burning vegetation that stimulate germination of many fire-following plants[13,14]. KARs, or more accurately putative KAR metabolites, signal through the α/β-hydrolase receptor KARRIKIN INSENSITIVE 2 (KAI2). Upon activation, KAI2 interacts with the SCF-type E3 ubiquitin ligase complex through the F-box protein MORE AXILLARY GROWTH 2 (MAX2) to target SMAX1 for polyubiquitylation and subsequent proteasomal degradation (summarized in refs. [14,15]). KAI2 is also thought to perceive (an) unknown endogenous signal(s), tentatively called KAI2 ligand(s) (KL)[14,16]. AM colonization of roots is absent or very reduced in *kai2* and *max2* mutants of the grasses *Oryza sativa* (rice), *Hordeum vulgare* (barley) and *Brachypodium distachyon*; the solanaceous species *Petunia hybrida*; and the legume *Medicago truncatula*[17–20]. Conversely, *smax1* mutants of rice show increased root colonization by AM fungi. This phenotype may be due to increased expression of strigolactone biosynthesis genes resulting in increased strigolactone exudation, as well as increased expression of Myc factor receptor (LysM-RLK), and several Common Sym genes[12,19]. Thus, SMAX1 seems to act as a gatekeeper for symbiosis that may be modulated by internal and/or external signals.

The *smax1* mutant also shows increased expression of *1-aminocyclopropane-1-carboxylic acid synthase 7* (*ACS7*), which encodes a central enzyme in ethylene biosynthesis. ACS7 produces the ethylene precursor 1-aminocyclopropane-1-carboxylic acid (ACC), leading to increased ethylene production in *smax1*[21,22]. This observation is paradoxical, as pharmacological application of ethylene and its biosynthetic precursors negatively impacts root colonization by AMF in several species[23–28]. Furthermore, treating an ethylene over-producing mutant (*epi*) in tomato with the ethylene biosynthesis inhibitor aminoethoxyvinylglycine (AVG) restores AM colonization to wild-type levels[29]. Loss-of-function mutations of *ETHYLENE INSENSITIVE PROTEIN 2* (*EIN2*), a central regulator of ethylene signaling, causes ethylene insensitivity[30], and the *Medicago truncatula sickle* (*ein2*) mutant promotes early stages of root colonization[17]. This negative effect of ethylene on AM colonization is difficult to reconcile with the increased colonization level seen in *smax1* roots, which overproduce ethylene[21] – unless *SMAX1* is required for the suppressive effect of ethylene.

The negative effect of ethylene on AM has been known for several decades[23], and it has been proposed that ethylene suppresses AM through induction of defense responses[31]. However, the molecular mechanisms underlying the negative effect of ethylene on AM remained elusive. Here we investigate the molecular events underpinning ethylene effects on AM using the model legume *Lotus japonicus* and the model AMF *Rhizophagus irregularis*. We report that ethylene acts through SMAX1 to suppress genes required for AM development.

## Results

### Ethylene inhibits *L. japonicus* root colonization by AMF
As ethylene signaling has not been studied in AM of *L. japonicus* before, we first examined the concentration-dependent and temporal effects of ethylene on root colonization of this plant by the model AMF

*Rhizophagus irregularis*. 1-aminocyclopropane-1-carboxylic acid (ACC) and ethephon are two commonly used ethylene precursors. ACC is a natural precursor that is converted to ethylene by plant ACC oxidase but can also have ethylene-independent signaling roles[32]. Ethephon is a synthetic precursor that releases ethylene and phosphate upon its spontaneous decomposition. Increased phosphate fertilization can suppress root length colonization, including in *L. japonicus*, in a dose-dependent manner[11]; thus, phosphate release could potentially obscure the effect of ethylene produced by ethephon. This led us to use both ACC and ethephon to robustly assess the impact of ethylene on *L. japonicus* root colonization by AMF.

Treatment with 200 μM ACC reduced total root colonization from ~70% to ~22%, and 500 μM ACC further suppressed the formation of arbuscules and vesicles (Fig. 1A). Similarly, total root colonization was reduced by two-thirds by 10 μM ethephon and abolished almost entirely by 100 μM ethephon (Fig. 1B). In a time-course with 200 μM ACC treatment, colonization was significantly inhibited within 17 days post inoculation (dpi) (Supplementary Fig. 1A). This suggests that ACC acts at early stages of AM establishment. A closer look at colonization units in roots treated with 200 μM ACC or 100 μM ethephon revealed multiple aborted hyphopodia and unusually short patches of colonization. These unsuccessful colonization patches were marked by the occurrence of septate hyphae and abnormally swollen hyphopodia (Supplementary Fig. 1B), which are considered hallmarks of aborted colonization events.

To test whether endogenous ethylene affects AM formation, we treated plants with the ethylene biosynthesis inhibitor AVG. Total colonization, as well as colonization with arbuscules, was increased by 10 μM AVG treatment (Fig. 1C). We also examined how AM colonization was regulated by *EIN2*, which acts in the signal transduction between ethylene perception and downstream transcriptional responses[33]. The *L. japonicus* genome contains two *EIN2* paralogs, *EIN2A* and *EIN2B*[34]. Loss of *EIN2B* caused an increase in root length colonization; in contrast, as previously reported[35], the *ein2a* mutant had no effect on colonization either on its own or in combination with *ein2b* (Fig. 1D). Furthermore, root colonization of *ein2a ein2b* was unaffected by 200 μM ACC and showed only a very small decrease in response to 100 μM ethephon (Fig. 1E, F). The weak effect of ethephon on root colonization of *ein2a ein2b* might be related to the release of phosphate by ethephon, but regardless contrasts sharply with the ethephon response of wild-type plants. Therefore, the effects of ACC and ethephon on colonization are a consequence of EIN2-mediated ethylene signaling specifically. Altogether these data demonstrate that ethylene suppresses root colonization by AMF in *L. japonicus*.

We noted that high concentrations of ACC and ethephon were required to suppress AM in sand-grown *L. japonicus*. We hypothesized that this could be caused by inefficient availability of these compounds in sand and/or due to the escape of volatile ethylene gas. Therefore, we examined whether ethylene-producing treatments would be more potent in aerated hydroponic culture[36]. Indeed, in hydroponic conditions, ACC and ethephon suppressed total root length colonization of *L. japonicus* at much lower concentrations (20 μM), and only 2 μM AVG was sufficient to promote colonization (Supplementary Fig. 2A, B). We observed similar results in rice plants grown hydroponically. While 10 μM AVG promoted AM in rice roots, 10 μM ethephon strongly suppressed colonization (Supplementary Fig. 3A, B). Thus, in hydroponic growth conditions, AM colonization is more sensitive to chemicals that perturb ethylene abundance.

### Ethylene signaling inhibits AM development in the epidermis
Since AMF on the surface of ACC- and ethephon-treated roots displayed aberrant hyphopodia that did not initiate root entry, we hypothesized that ethylene acts primarily in the root epidermis to limit colonization. To test this hypothesis, we expressed a hypermorphic version (version with increased activity of the gene product) of EIN2B

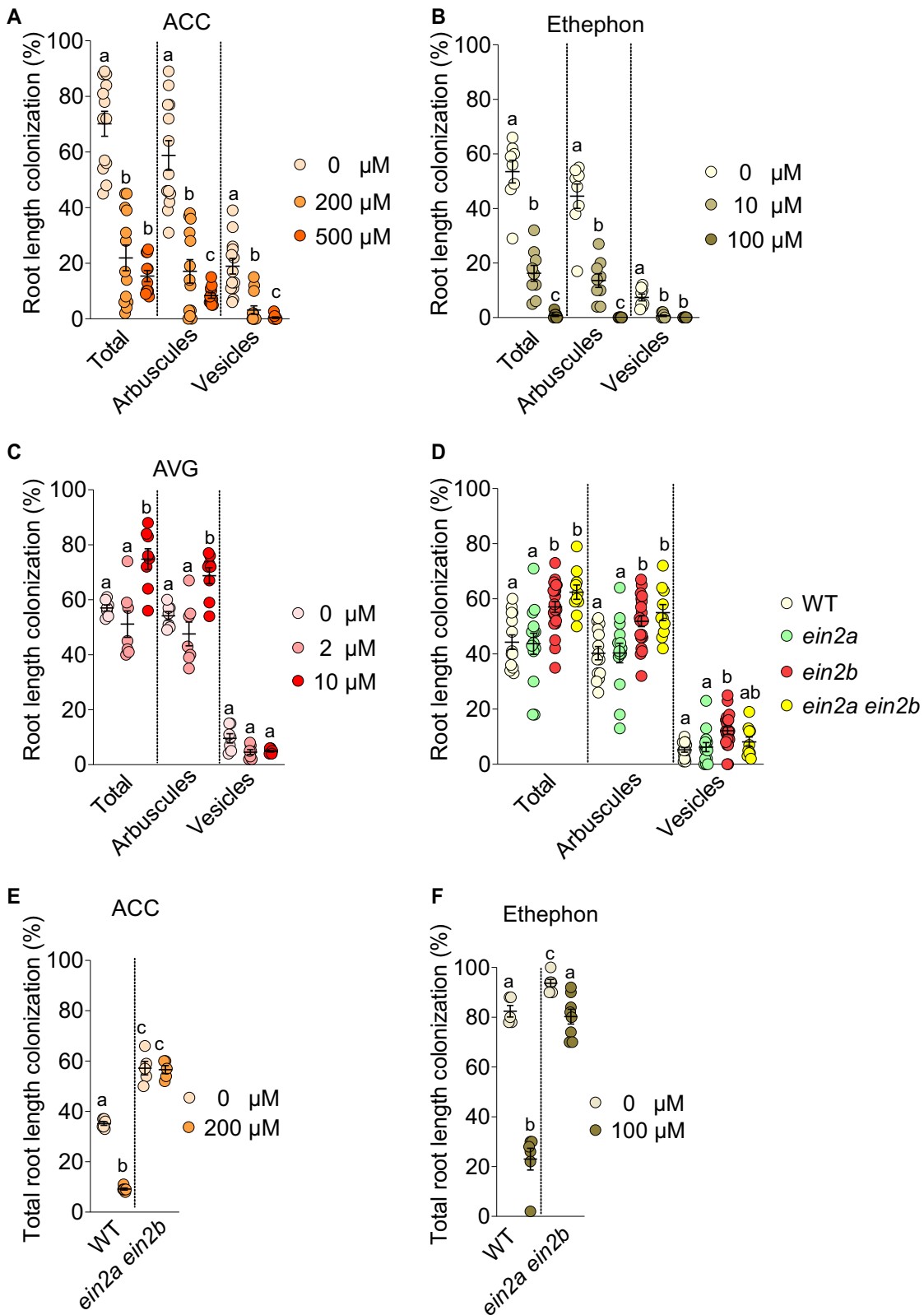

(EIN2-CEND) in the *ein2a ein2b* double mutant under the control of different root tissue-specific promoters. After ethylene perception, EIN2, which is localized to the ER membrane, is cleaved into two parts. The cytosolic EIN2 carboxyl end (CEND), which contains a nuclear localization signal, translocates to the nucleus to activate downstream ethylene responses[37]. Constitutive expression of CEND is sufficient for the activation of ethylene responses[30]. We expressed the coding

sequence of *EIN2B-CEND*, encoding amino acids 477-1308 of EIN2B, under the control of a constitutive ubiquitin promoter, an epidermis-specific (p*Epidermis* (p*Epi308* in[38])) promoter, and a cortex-specific promoter (p*Cortex*[39]). We also tested two AM-induced promoters that are either expressed 1) along the whole path of colonization starting in an epidermal cell under the hyphopodium and ending in arbuscule-containing cells (p*SbTM1*[40]) or 2) specifically in arbuscule-containing

**Fig. 1 | Manipulation of ethylene signaling affects AM in *Lotus japonicus*.** Percent root length colonization (RLC) (**A**–**C**) of wild-type *L. japonicus* at 4 wpi with *R. irregularis* (AMF) treated with ethylene precursors, 1-aminocyclopropane-1-carboxylic acid (ACC) or ethephon or with the ethylene biosynthesis inhibitor, aminoethoxyvinylglycine (AVG) at the indicated concentrations. **D** Percent RLC of *ein2* single and double mutants at 4 wpi. **E**, **F** Percent total RLC of *ein2* double mutants co-cultured with AMF for 4 (**E**) or 6 (**F**) weeks, and treated with ACC or ethephon at indicated concentrations. Experiments in **A**–**D** were performed three times independently with similar results. **A**–**D**; Kruskal-Wallis test [for (**A**) N = 13 for 0 μM and 200 μM ACC, 11 for 500 μM ACC, Kruskal-Wallis H statistic = 82.62 for total, 23.91 for arbuscules, 23.40 for vesicles; for (**B**) N = 8 for 0 μM and 100 μM Ethephon, 9 for 10 μM, Kruskal-Wallis H statistic = 67.87 for total, 21.14 for

arbuscules, 19.57 for vesicles; for (**C**) N = 7 for 0 μM and 2 μM AVG, 8 for 10 μM AVG, Kruskal-Wallis H statistic = 54.35 for total, 11.22 for arbuscules, 5.972 for vesicles; for (**D**) N = 14 for WT and *ein2a-2*, 21 for *ein2b-1*, 10 for *ein2a ein2b*, Kruskal-Wallis H statistic = 137.4 for total, 18.10 for arbuscules, 14.13 for vesicles] with Dunn's posthoc comparison. Different letters indicate statistical differences between treatments or genotypes. **E**, **F** Welch and Brown-Forsythe one-way ANOVA with Games-Howell's multiple comparisons test [for **E** N = 6 for WT, 5 for *ein2a ein2b*, F* (DFn, DFd) = 213.4 (3.000, 7.542) & W (DFn, DFd) = 538.8 (3.000, 8.257) and for (**F**) N = 5 for 0 μM ethephon/WT, 6 for 100 μM ethephon/WT and 0 μM ethephon/*ein2a ein2b*, 8 for 100 μM ethephon/*ein2a ein2b*, F* (DFn, DFd) = 108.8 (3.000, 14.06) & W (DFn, DFd) = 70.56 (3.000, 10.74)]. Different letters indicate statistical differences between treatments or genotypes.

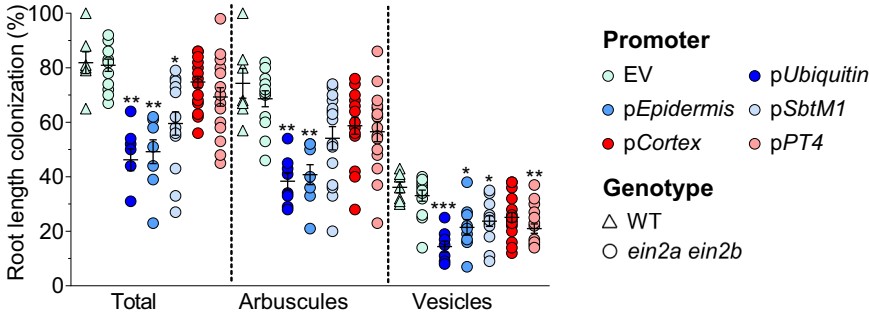

**Fig. 2 | Ethylene signaling suppresses AM majorly in the epidermis.** Percent root length colonization of wild-type hairy roots transformed with empty vector (EV), and of *ein2a-2 ein2b-1* hairy roots transformed with EV, or with the carboxy terminus of *EIN2B* expressed under the control of the indicated promoters. Statistics: Individual data-points and mean ± SE are shown. N = 7 for EV/WT, 13 for EV/*ein2a*

*ein2b*, 8 for p*Ubiquitin*, 9 for p*Epidermis*, 15 for p*SbtM1*, 18 for p*Cortex*, 17 for p*PT4*; Kruskal-Wallis test (Kruskal-Wallis H statistic = 44.55 for total, 33.90 for arbuscules, 34.63 for vesicles) with Dunn's posthoc comparison. Asterisks denote significance: * $p \le 0.05$; ** $p \le 0.01$; *** $p \le 0.001$.

cells (p*PT4*[41]). Total root length colonization, arbuscules, and vesicles were significantly decreased when *EIN2B-CEND* was expressed under the control of p*Ubi* as well as p*Epidermis*, but not p*Cortex* (Fig. 2). In addition, p*SbtM1*-driven (but not p*PT4*-driven) expression of *EIN2B-CEND* caused a significant, although lesser decrease in root colonization. All three promoters (p*Ubi*, p*Epidermis* and p*SbtM1*) have in common that they are active in the epidermis, although p*SbtM1* is only expressed under hyphopodia and not the whole epidermis, likely explaining the weaker effect on colonization. Together this suggests that ethylene signaling in the root epidermis inhibits fungal entry into the root, consistent with the abnormal, and unsuccessful hyphopodia observed on ACC- and ethephon-treated roots[29] (Supplementary Fig. 1B).

## Ethylene inhibits the expression of AM-relevant genes

To elucidate molecular mechanisms underlying the effect of ethylene on AM, we performed RNA-sequencing of *Rhizophagus irregularis*-colonized (AM) and mock-inoculated (Mock) roots of wild type treated with solvent or 20 μM ethephon, and *ein2a ein2b* double mutants treated with solvent (Supplementary Data 1 and 2, Supplementary Fig. 4A). To ensure treatments were homogeneous and to facilitate quick harvesting of clean roots in large amounts, we co-cultivated plants with *R. irregularis* in an aerated hydroponic growth system[36]. With respect to the number of differentially expressed genes (DEGs, cut-off: absolute($\log_2$Fold-Change) ≥ 0.59 and adjusted *p*-value ≤ 0.01) AM roots were less responsive to ethephon than Mock roots, while *ein2a ein2b* roots responded to AM with more DEGs than wild type (Supplementary Fig. 4B). In a principal component analysis, the transcriptomes separated along the Principal Component 1 (42% variation) according to AM vs. Mock status, and on the Principal Component 2 (22% variation) by ethylene signaling intensity, with ethylene-insensitive *ein2a ein2b* at one extreme and ethephon-treated Mock

wild type at the other (Supplementary Fig. 4C). Ethephon treatment reduced the variation between AM and Mock wild-type transcriptomes, implying ethylene reduced transcriptional responses to AM and/or reduced AM colonization.

We investigated whether ethylene affects the expression of genes that potentially have roles in AM development. To this end, we identified DEGs that are induced by AM colonization and compared these "AM-induced DEGs" to DEGs that are downregulated in Mock wild-type roots after ethephon treatment (Fig. 3A, Supplementary Data 3). (n.b. We isolated the variables in this manner because the comparison of ethephon-treated AM wild-type roots to solvent-treated AM wild-type roots confounds the effects of ethylene, reduced AM colonization, and interactions between ethylene and AM-responsive gene expression). Wild-type and *ein2a ein2b* roots shared 767 AM-induced DEGs. Among these, 209 (27.2%) genes were also downregulated by ethephon in wild-type Mock roots. This indicates that ethylene treatment inhibits the expression of many AM-induced genes even in the absence of AM fungi. Notably, 14 of the 58 AM-induced DEGs that were specific to wild-type roots were not identified as AM-induced in *ein2a ein2b* due to having high expression in *ein2a ein2b* already in the absence of AM. The increased number of AM-induced DEGs in *ein2a ein2b* roots compared to wild type (3154 vs 1876, respectively) further indicates that endogenous ethylene restricts AM-responsive gene expression.

GO term enrichment analysis of DEGs shared between any two conditions (Sets I, II, III) and all three conditions (Set IV) (Fig. 3A), showed that genes affected by ethephon treatment and by AM in *ein2a ein2b* (Set I, 716 genes) displayed enrichment for terms such as "cellular response to phosphate starvation", "gibberellic acid mediated signaling pathway", and "cell wall and carbohydrate related processes" (Fig. 3B; Supplementary Data 3), indicating some responses to ethylene described in *Arabidopsis thaliana* are conserved in *L. japonicus* and that ethylene signaling affects phosphate starvation responses,

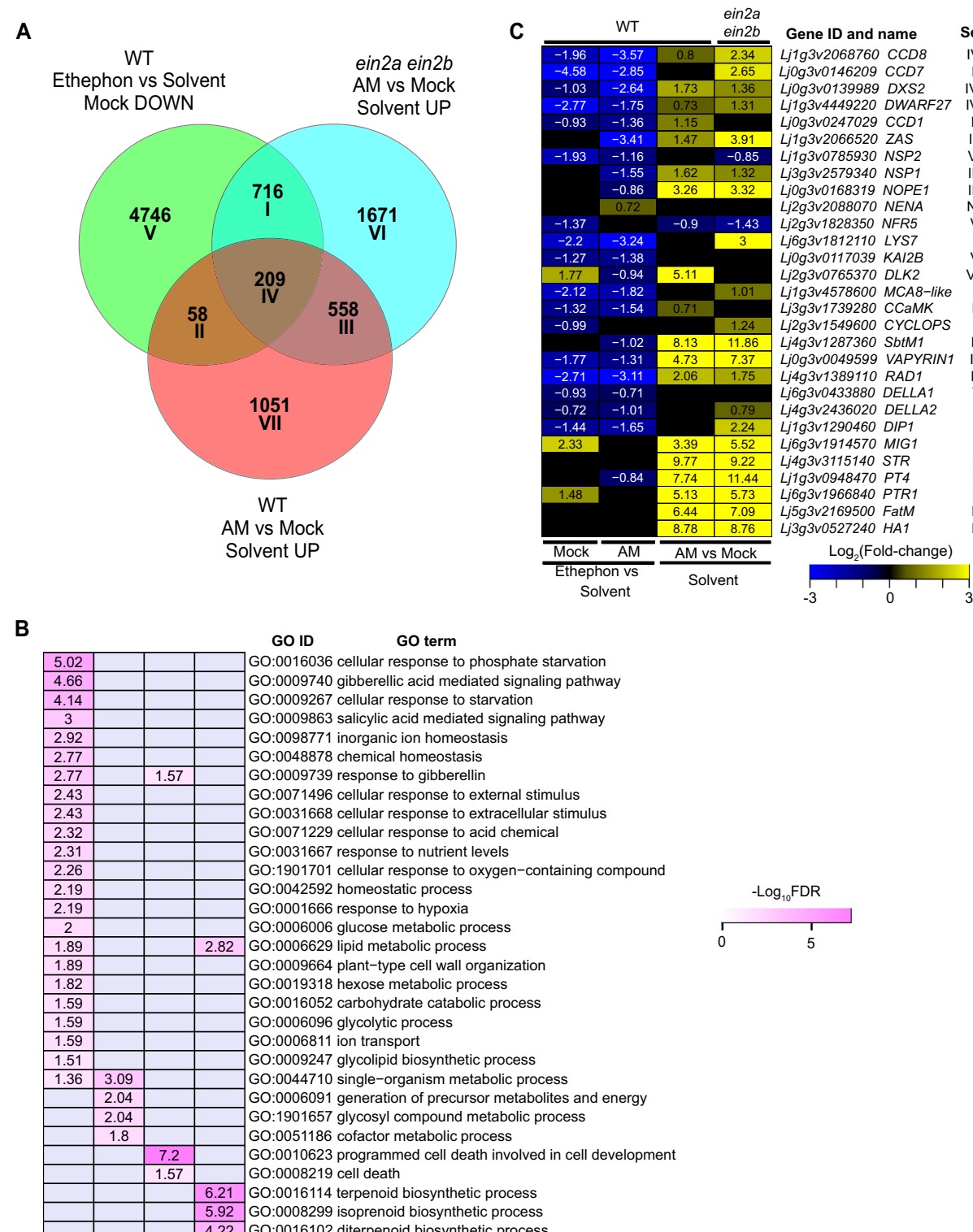

which strongly influence AM[11,42–44]. Furthermore, genes potentially related to early stages of root entry and hyphal progression through cell walls seem to be affected by ethylene. Two-hundred and nine DEGs, which overlapped for all conditions (Set IV), were enriched in terms related to secondary metabolism such as "terpenoid biosynthesis", "isoprenoid biosynthesis", "gibberellin metabolism" and "lipid biosynthetic process" which include genes that may be implicated in

AM symbiosis (e.g., for feeding the fungus with lipids at the arbuscule[45], or regulation of AM by gibberellin[46]).

We prepared an 'AM genelist' for *L. japonicus* (Supplementary Data 4) composed of 156 genes that 1) have been shown to be functionally relevant for AM symbiosis by forward or reverse genetics, 2) are conserved in genomes of AM-competent plants[47], 3) are induced in AM roots across several plant species suggesting a conserved and

**Fig. 3 | Expression of genes with functional importance for AM development is suppressed by ethylene. A** Venn diagram showing overlap of DEGs with decreased transcript accumulation in ethephon treated wild type *L. japonicus* roots and DEGs with increased transcript accumulation in solvent-treated AM roots of wild type and *ein2a ein2b*. **B** Gene ontology enrichment for Venn sets I–IV from (**A**). The values within the cells represent the logarithm of the FDR obtained for the enrichment of specific gene ontology terms. Darker colors indicate stronger enrichment. **C** Log₂Fold-change heatmap for genes previously reported to be functionally

required in AM and displaying significantly reduced expression in response to ethephon or increased expression in the *ein2a ein2b* mutant. Cells with numbers represent log₂Fold-change with significant difference to the respective comparator (adjusted *p*-value ≤ 0.05). Black cells represent no significant difference. 'Set' indicates the gene set in (**A**). To identify differentially expressed genes between groups, a two-sided exact test in edgeR, was used, accounting for both up- and down-regulated genes. Adjusted *p*-values were calculated using the Benjamini-Hochberg procedure to control for multiple comparisons.

important function in AM, and/or 4) belong to functional groups that are potentially implicated in AM, such as LysM receptor-like kinases, GRAS proteins, or ABC transporters. Forty-eight of the 156 genes (31%) in the AM genelist were downregulated by ethephon treatment in Mock wild-type roots, and 14 of these genes (9%) were upregulated by disrupted ethylene signaling (i.e., in *ein2a ein2b*) (Supplementary Fig. 5A). Therefore, more than one-third of these genes that may be involved in AM symbiosis are regulated by ethylene. We observed highly similar effects on differential gene expression when ethylene signaling was perturbed in AM-colonized plants (Supplementary Fig. 5B, Supplementary Data 6). Notably, 17 of the ethylene-responsive genes in the AM genelist have been genetically shown to be required for proper AM development. Strigolactone biosynthesis genes (*D27*, *CCD7*, *CCD8*, and a gene encoding one of their transcriptional regulators, *NSP2*[19]), Myc and Nod factor receptor *LysM-RLK* genes (*LYS7*[48,49], *NFR5*[50]), common symbiosis genes (*MCA8-like*, *CCaMK*, *CYCLOPS*, *DELLA*, *Vapyrin*, reviewed in ref. [10]), and the KAR/KL receptor gene *KAI2B*[51,52] showed reduced expression specifically upon ethephon treatment. In addition, expression of the KAR/KL receptor gene *KAI2A* and the canonical KAR/KL-response gene *DLK2*[52,53] was significantly increased in *ein2a ein2b*, suggesting that KAR/KL signaling is increased in ethylene-insensitive roots and decreased upon ethylene increase (Supplementary Fig. 5A, Supplementary Data 5).

To identify patterns of coregulation, we performed hierarchical clustering of DEGs that responded to ethephon in AM and Mock wild-type roots and DEGs that responded to AM in wild-type and *ein2a ein2b* roots. Among the 41 genes from the AM genelist with known functions in AM symbiosis, 25 were placed in DEG clusters 3 and 4, which are suppressed in response to ethephon treatment and induced in AM roots (Supplementary Fig. 6; Fig. 3C; Supplementary Data 7). This group includes genes involved in apocarotenoid biosynthesis (*DXS2*, *CCD1*, *ZAS*); strigolactone biosynthesis (*NSP1*, *DWARF27*, *CCD7*, *CCD8*), Myc factor perception (*LYS7*), (N-acetylglucosamine) transport of an exuded compound activating AM fungi (*NOPE1*), KAR/KL signaling (*KAI2B*), nuclear calcium spiking (*MCA8-like*), transcriptional regulation (*CCaMK*, *CYCLOPS*) and intraradical progression and early arbuscule development (*SbtM1*, *VAPYRIN1*, *RAD1*, *DELLA1 & 2*, *DIP1*)[10,48,54–56]. Interestingly, AM-relevant genes involved in nutrient exchange and arbuscule branching and maintenance (*MIG1*, *STR*, *PT4*, *NPF4.5*, *FatM*, *HA1*) were induced upon AM in both wild-type or *ein2a ein2b* roots but were not repressed after ethephon treatment in either the absence or presence of AM (Fig. 3C). This supports the notion that ethylene mainly acts during the early stages of AM colonization in the root epidermis rather than during later stages of arbuscule formation (Fig. 2, Supplementary Fig. 1B).

To identify new genes that might have roles in AM establishment, we examined the expression profiles of uncharacterized genes that fall into functional categories likely to be associated with AM symbiosis. These categories included carotenoid cleavage dioxygenases, which are potentially involved in strigolactone/apocarotenoid metabolism; strigolactone signaling proteins; ABC transporters, which might mobilize strigolactones, related signals or other signaling or nutritional compounds; LysM-RLKs; GRAS proteins; and gibberellin metabolism enzymes. *CCD-like (Lj1g3V2067530)*, *SMXL8*, *PDR1*, *PDR1-like (Lj5g3v0540130)*, *PDR2-like (Lj0g3v0252699)*, *LYS15*, *LYS17*, *DELLA3*,

*RAD1-like (Lj3g3v1486970)*, *SCL13-like (Lj6g3v1752890)*, *SCL26-like (Lj0g3v0017249)*, *Lj1g3v4850500 (GRAS)*, *Lj6g3v1945170 (GA2OX)* and *Lj0g3v0109249 (GA3OX1)* displayed reduced transcript accumulation upon ethephon treatment and increased expression in AM roots treated with solvent (Supplementary Fig. 7). Along with uncharacterized genes in DEG clusters 3 and 4, these genes may be promising candidates for future genetic studies of AM symbiosis, for example to identify novel LysM receptors involved in symbiotic signaling (Supplementary Figs. 6 and 7).

To confirm that ethylene causes downregulation of AM-relevant transcripts prior to AM colonization, we treated non-colonized plants grown on Petri dishes with ACC and performed RT-qPCR for an ethylene biosynthesis gene, three strigolactone biosynthesis genes, three *LysM-RLK* genes, two Common Sym genes (*CCaMK*, *CYCLOPS*), three *DELLA* genes, and a defense response gene (Supplementary Fig. 8). Supporting the inhibitory effect of ethephon on expression of genes relevant for AM symbiosis, the transcript accumulation of all but two of these genes was reduced upon ACC treatment. Only two genes used as positive control for the effect of ethylene *ACC OXIDASE 2 (ACO2)*, which is involved in ethylene biosynthesis, and *Lj0g3v0286359*, *PATHOGENESIS-RELATED PROTEIN 10 (PR10)*, which is involved in defense response, were induced by ACC treatment. *PR10* expression was also reduced by AVG treatment, validating its positive regulation by ethylene. Altogether these data demonstrate that ethylene represses the expression of many important genes involved in AM development.

## Treatment with a strigolactone analog and ectopic expression of CCaMK revert the ethylene effect on root colonization

Strigolactones are important activators of AMF spore germination and hyphal branching[3,57]. We observed that the strigolactone biosynthesis genes *D27*, *CCD7*, and *CCD8* were repressed by ethephon and ACC treatment, and *CCD7* further showed increased expression in *ein2a ein2b* compared to wild type in the presence of AM (Supplementary Fig. 5). This led us to hypothesize that ethylene may reduce AM colonization by inhibiting strigolactone biosynthesis. To test this idea, we co-treated roots with ACC and low concentrations (5 or 10 nM) of a synthetic strigolactone analog, GR24[5DS], which is sufficient to activate the fungus but does not induce developmental responses in plants[53,57]. Exogenous application of GR24[5DS] at both concentrations counteracted the inhibitory effect of ACC treatment on root length colonization (Fig. 4A). Therefore, reduced strigolactone biosynthesis (and exudation) is a highly likely, but not necessarily exclusive, explanation for the inhibitory effects of ethylene on AM symbiosis.

The calcium-calmodulin-dependent kinase CCaMK is a central regulator of root symbiosis. Gain-of-function CCaMK (CCaMK[T265D] or lacking the C-terminal inhibitory domain CCaMK[1-314]) can bypass the requirement for perception of the fungus in the root epidermis and the induction of Ca²⁺-spiking[58,59]. As *CCaMK* expression was reduced by increased ethylene (Fig. 3C), we tested if ectopic expression of *CCaMK* driven by a ubiquitin promoter could overcome the ethylene-mediated suppression of root colonization and expression of AM marker genes. In contrast to the empty vector control, colonization of hairy roots

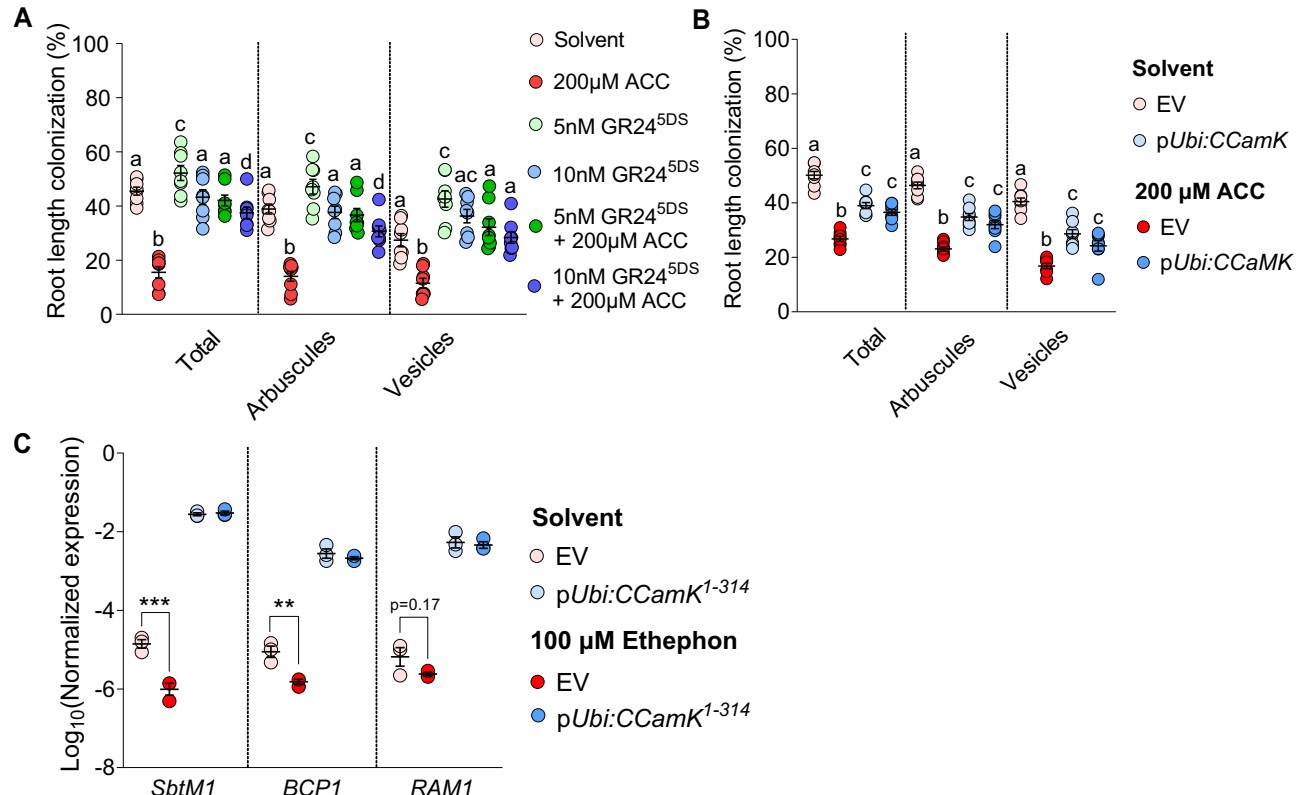

**Fig. 4 | Supplementation with GR24^{5DS} and ectopic expression of CCaMK restore ethylene-suppressed root colonization. A** Percent root length colonization, of wild-type *L. japonicus*, with *R. irregularis* at 4 wpi and treatment with the indicated chemicals at indicated concentrations. **B** Percent root length colonization of *L. japonicus* hairy roots transformed with empty vector (EV), p*Ubi:CCaMK*, with *R. irregularis* at 5 wpi treated with solvent (water) or ACC. **A**, **B** The experiments were repeated twice with similar results. **C** RT-qPCR-based expression of AM marker genes in absence of the fungus in *L. japonicus* hairy roots carrying the indicated transgenes. Expression of *SbtM1*, *BCP1* and *RAM1* was normalized to the housekeeping gene *Ubiquitin*. Statistics: (**A**) Individual data-points and mean ± SE (N = 8) are shown. Kruskal-Wallis test (Kruskal-Wallis H statistic = 19.80 total, 12.34 arbuscules, 0.9343 vesicles) with Dunn's posthoc comparison was used to assess significant differences between treatments for each AMF structure. **B**, **C** Two-way ANOVA with Tukey's multiple comparisons test was used to assess significant differences between treatments and genotypes. **B** Individual data-points and mean ± SE (N = 8) are shown. For total: interaction F (1, 28) = 85.81 (*P* < 0.0001), treatment F (1, 28) = 131.1 (*P* < 0.0001), genotype F (1, 28) = 0.4159 (*P* = 0.5242); for arbuscules: interaction F (1, 28) = 71.81 (*P* < 0.0001), treatment F (1, 28) = 116.0 (*P* < 0.0001), genotype F (1, 28) = 1.254 (*P* = 0.2723); for vesicles: interaction F (1, 28) = 42.46 (*P* < 0.0001), treatment F (1, 28) = 89.84 (*P* < 0.0001), genotype F (1, 28) = 2.263 (*P* = 0.1437). Letters indicate significant differences. **C** Individual data-points and mean ± SE (N = 3) are shown. For *SbtM1*: interaction F (1, 8) = 37.06 (*P* = 0.0003), treatment F (1, 8) = 32.72 (*P* = 0.0004), genotype F (1, 8) = 1592 (*P* < 0.0001); for *BCP1*: interaction F (1, 8) = 10.47 (*P* = 0.0119), treatment F (1, 8) = 19.61 (*P* = 0.0022), genotype F (1, 8) = 798.3 (*P* < 0.0001); for *RAM1*: interaction F (1, 8) = 1.618 (*P* = 0.2391), treatment F (1, 8) = 2.957 (*P* = 0.1238), genotype F (1, 8) = 454.7 (*P* < 0.0001). Asterisks indicate significance level: * *p* ≤ 0.05; ** *p* ≤ 0.01; *** *p* ≤ 0.001 **** *p* < 0.0001.

ectopically expressing *CCaMK* remained largely unaffected by ACC treatment (Fig. 4B). This result indicates that the suppression of *CCaMK* expression also contributes to the suppression of AM development by ethylene. Additionally, AM marker genes such as *SbtM1* and *BCP1* induced already during early stages of AM development, but not *RAM1*, involved in arbuscule development, were suppressed by ethephon treatment in hairy roots in the absence of AM. Ectopic expression of the dominant active CCaMK kinase domain (CCaMK^{1-314}) increased the expression of these genes and rendered their expression unresponsive to ethephon (Fig. 4C). Together this indicates that ethylene inhibits AM symbiosis through cumulative effects on the expression of AM-relevant genes, such as those engaged in strigolactone biosynthesis or intraradical accommodation of the fungus (Common Sym genes, represented here by *CCaMK*).

**Ethylene and SMAX1 affect overlapping genes**

This raised the question of how ethylene signaling leads to changes in the expression of genes associated with AM symbiosis. SMAX1 is a transcriptional regulator that is targeted for polyubiquitylation and degradation by the KAR/KL signaling complex, KAI2-SCF^{MAX2 60,61}. Consistent with a requirement for the KAR/KL receptor complex for

AM in rice (D14L-SCF^{D3}), rice *smax1* mutants have increased root colonization[12,51]. Rice *smax1* mutants also show increased expression of strigolactone biosynthesis genes and some Common Symbiosis genes in the absence of AM[12]. Because some of these genes overlap with those, we found to be suppressed by ethylene treatment in *L. japonicus*, we hypothesized that *SMAX1* expression or SMAX1 protein abundance may be linked to ethylene signaling in AM symbiosis.

To address this, we first examined the colonization of *L. japonicus smax1-2* and *smax1-3* mutant roots. Similar to rice[12], *L. japonicus smax1* mutants were more strongly colonized than wild type (Fig. 5A). To identify genes regulated by *SMAX1* in *L. japonicus*, we performed RNA-seq analysis of non-colonized *smax1-2* and *smax1-3* roots (Supplementary Fig. 9; Supplementary Data 8 and 9). We compared the DEGs with increased expression in both *smax1* mutants to the DEGs with reduced expression in ethephon-treated Mock wild-type roots and to the AM genelist (Fig. 5B, C). 720 of the 2279 (32%) DEGs with increased expression in *smax1* had decreased expression upon ethephon treatment. 22 of these genes were also present on the 156-member AM gene list. Of these, 9 genes have been genetically shown to function in AM (Fig. 5B). These include strigolactone biosynthetic genes (*CCD7*, *CCD8*[62,63]), apocarotenoid biosynthesis genes (*DXS2*[64], *CCD1*[55]), a Myc

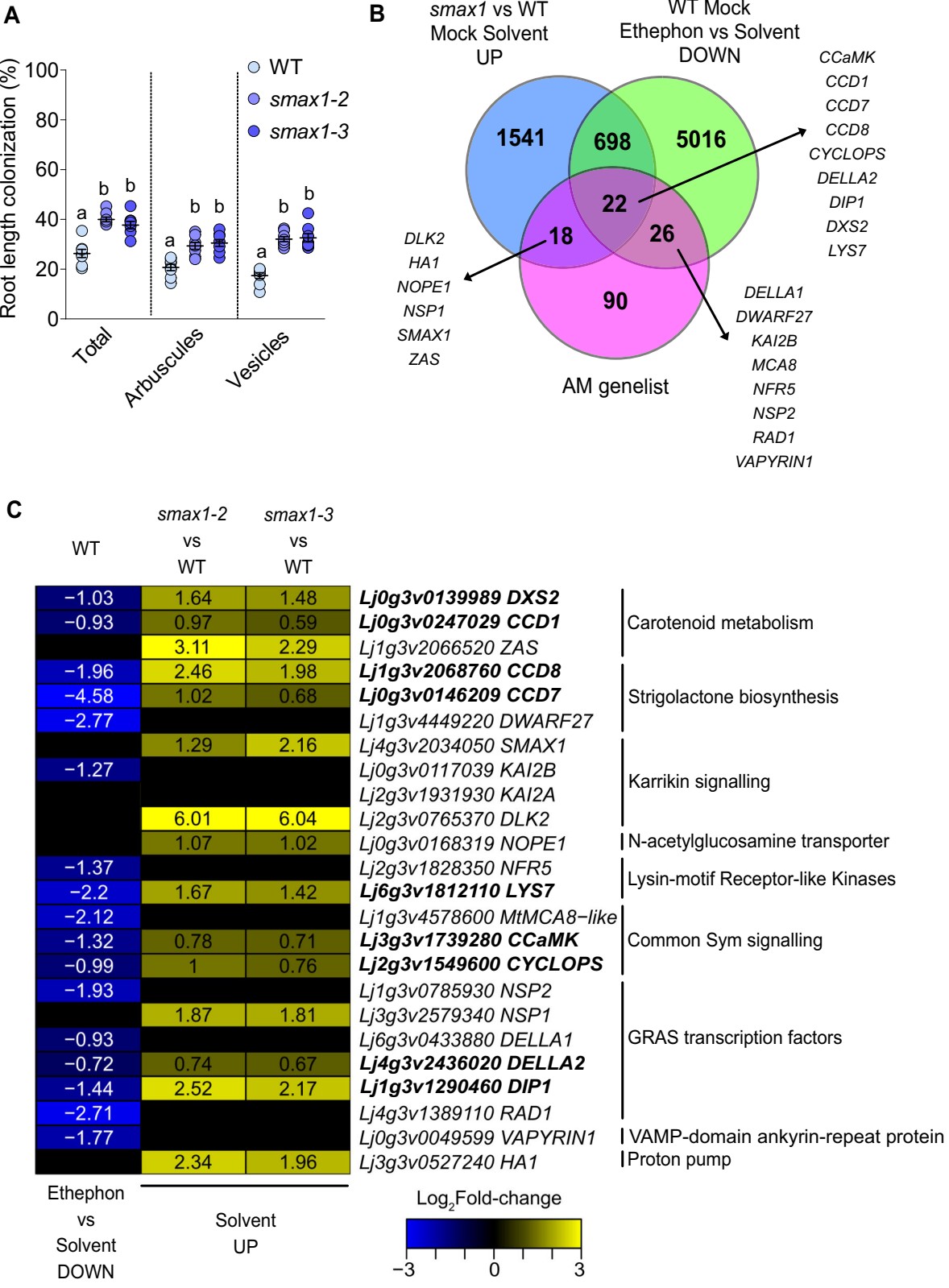

factor receptor gene (*LYS7*[148,49]), common Sym genes (*CYCLOPS*[65], *CCaMK*[58], *DELLA2*[9,46]), and an additional transcriptional regulator gene encoding an interactor of DELLA (*DIP1*)[66] (Fig. 5B, C, Supplementary Data 10). Many genes that have not yet been tested for roles in AM symbiosis showed opposite differential expression in *smax1* and ethephon-treated roots (698 genes in Fig. 4B, Supplementary Fig. 10). This included genes with predicted functions in gibberellin

biosynthesis, ABC transporters, and LysM-RLK or GRAS transcription factors. This demonstrates the regulation of a shared set of AM-relevant genes by ethylene and SMAX1.

**Ethylene signaling promotes SMAX1 accumulation**
To examine whether SMAX1 is required for the suppressive effect of ethylene on AM we inoculated *smax1* with *R. irregularis* and treated

**Fig. 5 | AM-relevant genes suppressed by ethylene are also suppressed by SMAX1. A** Percent root length colonization (RLC) of *L. japonicus* wild type, *smax1-2* and *smax1-3* mutants co-cultured with *R. irregularis* (AMF) for 4 weeks. Individual data-points and mean ± SE are shown. N = 8. Different letters indicate statistical differences between genotypes. Welch and Brown-Forsythe one-way ANOVA with Holm-Sidak's multiple comparisons test was performed [for total, F* (DFn, DFd) = 28.34 (2.000, 17.98) & W (DFn, DFd) = 25.48 (2.000, 13.03); for arbuscules, F* (DFn, DFd) = 16.03 (2.000, 20.95) & W (DFn, DFd) = 15.82 (2.000, 13.99); for vesicles, F* (DFn, DFd) = 43.31 (2.000, 17.58) & W (DFn, DFd) = 49.22 (2.000, 13.52)]. **B** Venn diagram showing the overlap of DEGs with increased expression in *smax1-2* and *smax1-3* roots in the absence of AMF with DEGs with reduced expression in ethephon-treated roots in the absence of AMF, and the AM

genelist. Names of genes with genetically determined roles in AM symbiosis are shown. **C** Log₂Fold-change heatmap for genes with genetically determined function in AM and displaying reduced expression after ethephon treatment or increased expression in the *smax1* mutants. Genes written in bold belong to the 22 genes in the overlap of all three gene sets in (**B**). Numbers in cells represent significantly different (adjusted *p*-value ≤ 0.05) log₂fold-change. Scale shows log₂fold-change with blue indicating decrease and yellow indicating increase. Black cells represent non-significant change. To identify differentially expressed genes between groups, a two-sided exact test in edgeR, was used, accounting for both up- and down-regulated genes. Adjusted *p*-values were calculated using the Benjamini-Hochberg procedure to control for multiple comparisons.

them with ACC. The mutants did not show any decrease in colonization upon ACC treatment (Fig. 6A, Supplementary Fig. 11A). In addition, expression of EIN2B-CEND in the root epidermis of *smax1* mutants hardly affected colonization, whereas it caused a strong suppression of colonization in the wild type (Fig. 6B). Furthermore, *ein2a ein2b smax1* triple mutants displayed root colonization levels that were similar to *smax1* single and *ein2a ein2b* double mutants, and not additive (Fig. 6C), indicating that *EIN2* and *SMAX1* act in the same pathway with respect to AM. We hypothesized that ethylene induces accumulation of SMAX1, which then leads to suppression of AM-relevant genes and AM colonization. Because *SMAX1* expression was not changed by ethephon treatment or the loss of ethylene signaling in *ein2a ein2b* (Fig. 6D), we tested whether ethylene affected the abundance of SMAX1 protein.

To observe SMAX1 accumulation in *L. japonicus* nuclei[21,67] we fused a GFP-tag to a SMAX1 C-terminal domain, SMAX1$_{D2}$, which is more easily detected than the full-length protein[67]. SMAX1$_{D2}$-GFP was expressed in *L. japonicus* wild-type and *ein2a ein2b* roots under the control of a constitutive ubiquitin promoter (p*Ubi:SMAX1$_{D2}$-GFP*), as an endogenous *SMAX1* promoter did not produce detectable amounts of the fusion protein in *L. japonicus* roots. We observed SMAX1$_{D2}$-GFP signal by confocal microscopy and quantified the intensity of GFP fluorescence relative to a p*35S:mCherry* control carried on the same T-DNA cassette (Fig. 6E, F; Supplementary Table 1). In solvent-treated wild-type and *ein2a ein2b* roots, SMAX1$_{D2}$-GFP accumulated strongly in only very few nuclei, and weakly or not at all in most nuclei. In contrast, upon ACC treatment of wild-type roots, all nuclei accumulated SMAX1$_{D2}$-GFP signal and the number of nuclei with strongly visible GFP signal increased. This was not the case for *ein2a ein2b* mutant roots, where SMAX1$_{D2}$-GFP abundance remained sparse. We confirmed the accumulation of SMAX1$_{D2}$-GFP after ACC treatment by Western Blot in *L. japonicus* hairy roots and were also able to detect full-length SMAX1 after ACC treatment in transiently transformed *Nicotiana benthamiana* leaves (Supplementary Fig. 11B). These observations suggest that ethylene via EIN2 causes accumulation of SMAX1, probably by acting at the level of translation or protein stability, as the use of the constitutive ubiquitin promoter makes transcriptional regulation by ethylene unlikely. Indirectly supporting this idea, expression of the canonical KAR/KL response gene *DLK2*, which is inhibited by SMAX1, showed a strong increase in *ein2a ein2b* in the absence of AM (Fig. 6C).

To understand whether the effect of ethylene on SMAX1 accumulation is associated with AM competence or can also be observed in species that cannot associate with AM fungi, we also treated *Arabidopsis* seedlings expressing tagged SMAX1$_{D2}$ with ACC. We performed a live luciferase assay of transgenic lines carrying p*UBQ10:SMAX1$_{D2}$-LUC* over 24 h treated with three different concentrations of ACC (Supplementary Fig. 11C). Strong luciferase activity was detected at the time of treatment, possibly due to treatment stress. Luciferase activity then decreased over time and plateaued at 8 h. Between 12 and 16 h, *Arabidopsis* seedlings treated with 1 or 10 μM ACC showed a 50% increase in luciferase activity, indicating SMAX1$_{D2}$-LUC accumulation, while plants treated with 0 or 100 μM ACC did not show this increase.

This indicates a time- and concentration-dependent accumulation of SMAX1 in response to ethylene. We also visualized the accumulation of SMAX1$_{D2}$-GFP expressed under the control of the *SMAX1* promoter in transgenic *Arabidopsis* roots by confocal microscopy after treatment with 10 μM ACC (Supplementary Fig. 11D). SMAX1$_{D2}$-GFP was not detectable in solvent-treated roots, but visibly accumulated in the nuclei of root tip cells after 14 h of ACC treatment. This suggests that ethylene-stimulated accumulation of SMAX1 protein could be a general phenomenon in plants.

We reasoned that if SMAX1 accumulation is responsible for the suppression of AM by ethylene, then co-treatment with KAR, which induces degradation of SMAX1 (summarized in refs. 15,68), should counteract the effect of ACC on AM. Indeed, 1 μM KAR₁ restored the colonization of *L. japonicus* roots treated with 200 μM ACC to the level of the solvent control (Fig. 6F). In addition, SMAX1$_{D2}$-GFP abundance in nuclei of *L. japonicus* hairy roots was reduced by KAR₁ in a *KAI2*-dependent manner (Supplementary Fig. 12, Supplementary Table 2). As expected, *kai2a kai2b* mutants showed higher accumulation of SMAX1$_{D2}$-GFP in nuclei of non-treated transgenic hairy roots, but surprisingly there was still a significant number of nuclei without signal. This suggested that either SMAX1 protein biosynthesis is adjusted or SMAX1 stability is regulated by additional factors. Furthermore, ACC treatment further increased SMAX1 abundance in *kai2a kai2b* roots indicating that promotion of SMAX1 accumulation by ethylene is independent of KAI2. It has recently been shown that SMAX1 can also be targeted by the strigolactone receptor D14 when plants are treated with osmotic stress or with 10 μM of the strigolactone analog GR24$^{SDS}$ [69]. To ensure that the small concentration of 5 nM GR24$^{SDS}$ intended to compensate for reduced strigolactone exudation upon ethylene precursor treatment, only activates the fungus (Fig. 4A) and does not lead to SMAX1 degradation, we co-treated SMAX1$_{D2}$-GFP expressing hairy roots with ACC and 5 nM GR24$^{SDS}$. The low concentration of GR24$^{SDS}$ did not influence SMAX1$_{D2}$-GFP accumulation caused by ACC treatment (Supplementary Fig. 13, Supplementary Table 3), indicating that, as expected, 5 nM GR24$^{SDS}$ only acts on the fungus and not on the plant SMAX1. Considering all data described above, we conclude that ethylene promotes the accumulation of SMAX1 protein in an EIN2-dependent manner. This represses the expression of AM-relevant genes, thereby preventing efficient AM development (Fig. 7).

## Discussion

The extent of arbuscular mycorrhiza development in roots is shaped by the physiological state of the plant[70]. Stresses such as high salinity, flooding, or shade can reduce root colonization[71–73]. A major regulator of stress mitigation is the gaseous hormone ethylene, which mediates responses to stressors such as flooding/hypoxia, cold, heat, drought, salt or heavy metals (summarized in ref. 28). Ethylene also suppresses AM development[23–27]. Therefore, although it has not been directly shown, it is conceivable that ethylene may be involved in mediating AM suppression by stress. The negative impact of ethylene on AM has been known for more than 40 years, but the molecular targets of ethylene

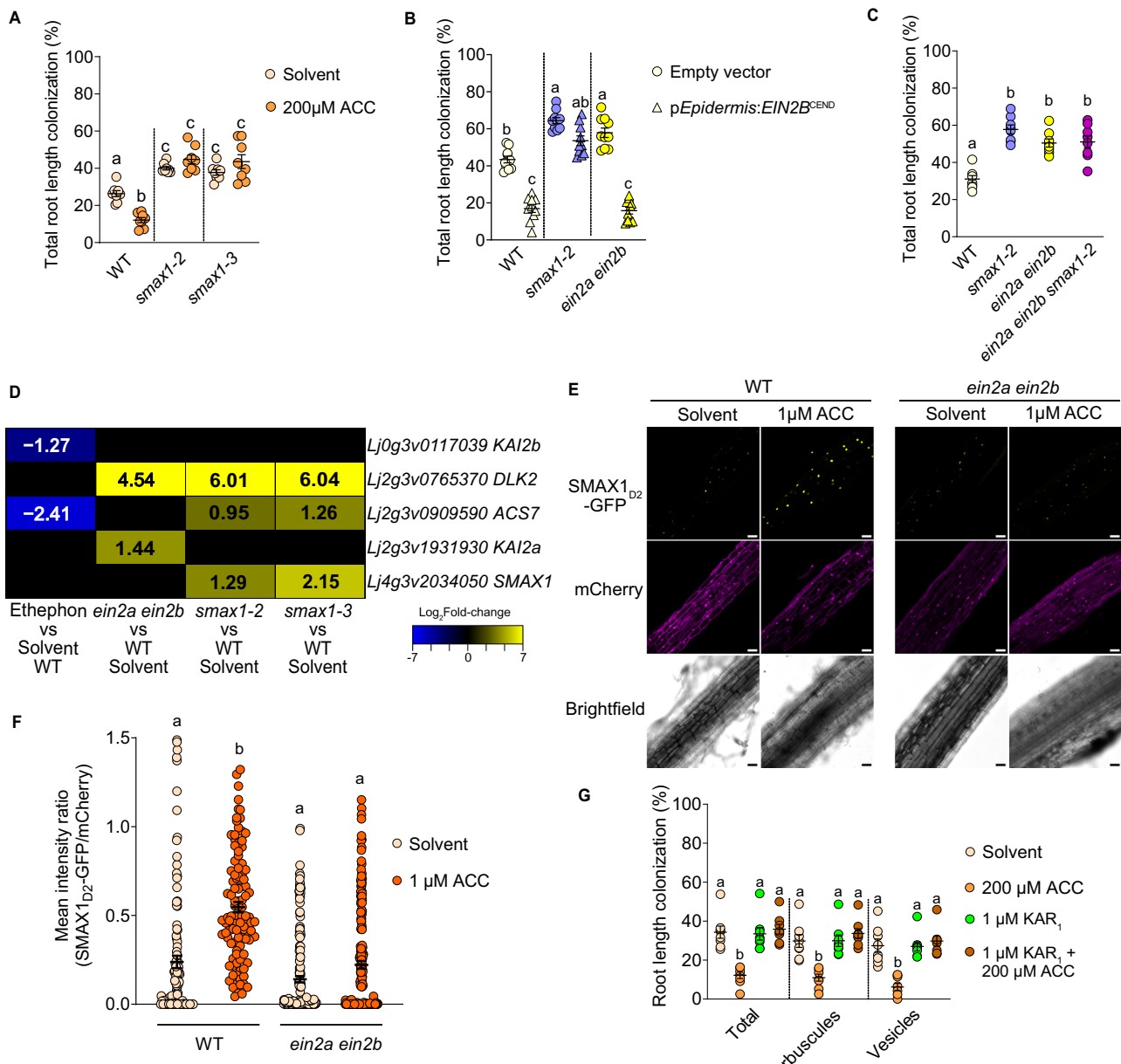

**Fig. 6 | Ethylene signaling promotes SMAX1 protein accumulation. A–C** Percent root length colonization of indicated *L. japonicus* genotypes with *R. irregularis* (AMF) after indicated treatments at 4 wpi (**A**, **B**) or 5 wpi (**C**). **D** Log$_2$Fold-change heatmap for indicated genes, genotypes and treatments. Numbers in cells represent significantly different (adjusted *p*-value ≤ 0.05) log$_2$fold-change. Black cells represent non-significant change. **E** Confocal microscopy images of *L. japonicus* wild-type and *ein2a ein2b* hairy roots expressing p*Ubi:SMAX1$_{D2}$-GFP* and a free p*35S:mCherry* transformation marker from the same T-DNA and treated for 24 hours with solvent (water) or 1 µM ACC. Scale bars = 50 µm. **F** Ratios of mean intensities of nuclear GFP signal to nuclear mCherry signal measured in confocal microscopy images in **E**. **G** Percent root length colonization of *L. japonicus* wild type at 4 wpi with *R. irregularis* treated with solvent (0.0075% methanol for KAR$_1$ and water for ACC), karrikin$_1$ (KAR$_1$), ACC or a combination of both. Statistics: **A** Individual data-points and mean ± SE (N = 8) are shown. Two-way ANOVA [interaction F (2, 42) = 14.17 (P < 0.0001), treatment F (1, 42) = 0.4810

(P = 0.4918), genotype F (2, 42) = 73.94 (P < 0.0001)] with Tukey's multiple comparisons test was used to assess significant differences between treatments and genotypes. **B** Individual data-points and mean ± SE (N = 8) are shown. Kruskal-Wallis test (Kruskal-Wallis H statistic = 19.56) with Dunn's posthoc comparison assessed significant differences between genotypes and/or treatments. **C** Individual data points and mean ± SE are shown. N = 10; Kruskal-Wallis test [H statistic = 49.01] with Dunn's posthoc comparison. **F** Individual data-points and mean ± SE (N = 111 for Solvent/WT, 107 for 1 µM ACC/WT, 177 for Solvent/*ein2a ein2b*, 184 for 1 µM ACC/ *ein2a ein2b*) are shown. Kruskal-Wallis test (Kruskal-Wallis H statistic = 115.0) with Dunn's posthoc comparison assessed significant differences between genotypes and treatments. **G** Individual data-points and mean ± SE (N = 8) are shown. Kruskal-Wallis test (Kruskal-Wallis H statistic = 17.97 for total, 18.54 for arbuscules, 18.31 for vesicles) with Dunn's posthoc comparison was used to assess significant differences between treatments for each indicated AMF structure. **A–C**, **F–G** Different letters indicate statistical differences.

signaling in the context of AM have remained elusive. Here we have shown that ethylene leads to suppression of genes with important roles in AM development by promoting accumulation of SMAX1, the proteolytic target of karrikin/KL signaling.

We confirm in *L. japonicus* that AM formation is reduced by treatment with the ethylene precursors ACC and ethephon, and that this depends on the central ethylene signaling component EIN2 (Fig. 1, Supplementary Figs. 1 and 2). We demonstrate that an increase in

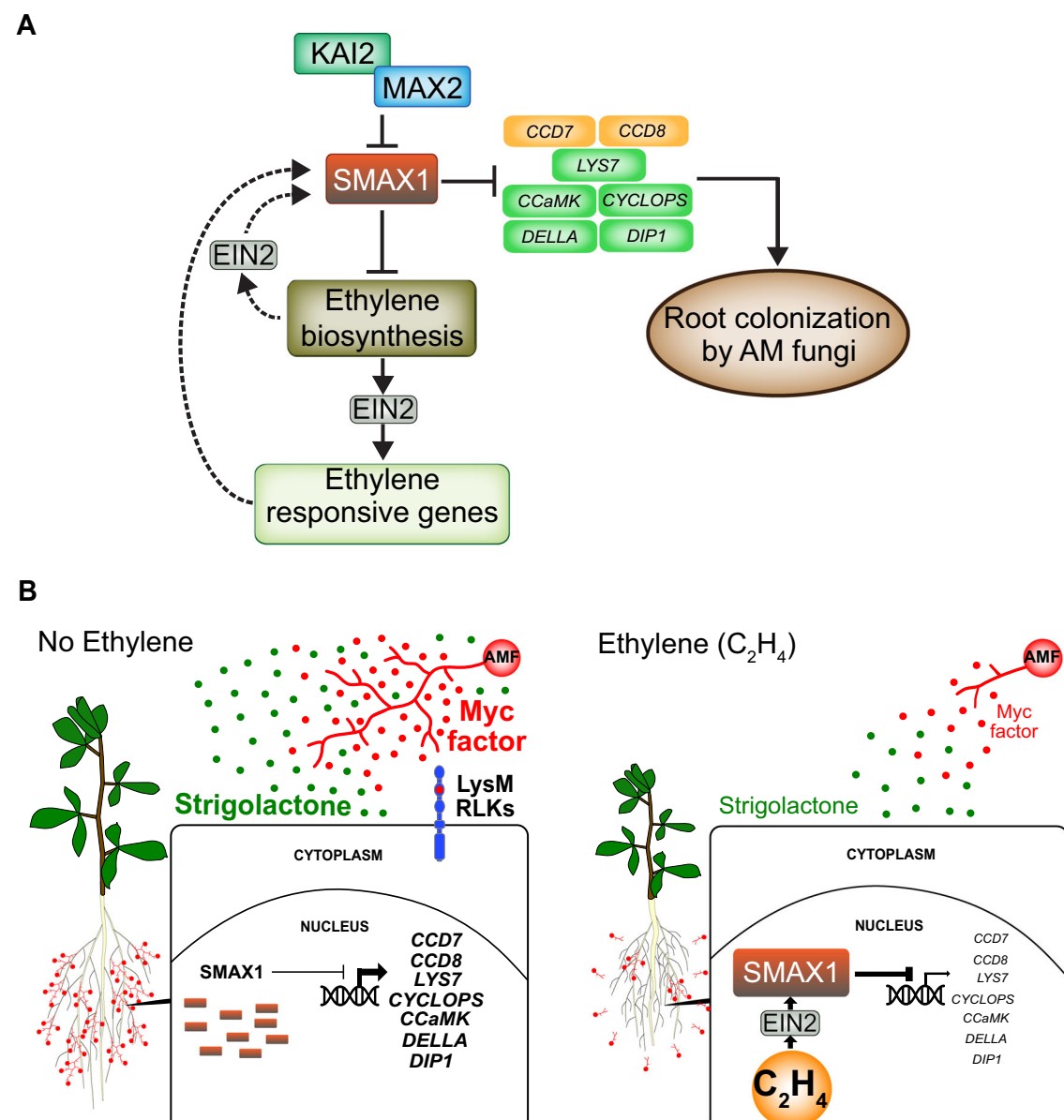

**Fig. 7 | Model for suppression of arbuscular mycorrhiza symbiosis by ethylene.**
**A** Upon perception of KAI2-ligand, KAI2 forms a complex with the SCF complex through the F-box protein MAX2 to ubiquitylate SMAX1 leading to its proteasomal degradation. This releases AM-relevant strigolactone biosynthesis genes (e.g., *CCD7*, *CCD8*), LysM receptors and common symbiosis genes (*LYS7*, *CCaMK*, *CYCLOPS*, *DELLA*), *DELLA INTERACTING PROTEIN 1* (*DIP1*) and other (AM-relevant) genes (not shown), from suppression allowing efficient AM development[12] (this work). Ethylene via EIN2 or via the product of ethylene-responsive genes (dashed arrows indicate hypothetical connections) promotes accumulation of SMAX1 and, therefore, suppression of the AM-relevant genes leading to suppression of AM. SMAX1 also acts as a negative regulator of an ethylene biosynthesis gene[21]. Thus,

SMAX1 degradation leads to increased ethylene biosynthesis, again causing accumulation of SMAX1[21]. This negative feedback loop may allow fine-tuning of root colonization by AMF in accordance with the stress level of the plant. **B** At low levels of ethylene and phosphate starvation[11] plants strongly express strigolactone biosynthesis genes, Myc factor receptor genes (LysM RLKs) as well as common symbiosis genes to stimulate the fungus with exuded strigolactone and to accommodate the fungus inside roots. At high levels of ethylene SMAX1 accumulates, leading to repression of strigolactone biosynthesis, Myc factor receptor and Common Symbiosis genes and as a consequence poor stimulation and intraradical accommodation of the fungus.

ethylene leads to reduced expression of a number of genes essential for AM development prior to colonization (i.e., in non-colonized roots), such as strigolactone biosynthetic genes required for fungal activation[4,57], LysM receptor-like kinase genes (*Lys7*) involved in the perception of fungal Myc factors[48,49], and common symbiosis genes (*CCaMK*, *CYCLOPS*) required for fungal intraradical accommodation[9,65,74] or arbuscule development (*DELLA*[46]). Mutations in these genes lead to strong AM developmental phenotypes. Thus, it is likely that their cumulative reduction in expression prevents efficient AM development upon ethylene increase.

For example, the AM phenotypes of *ccamk* and *cyclops* mutants of *L. japonicus* are characterized by bloated hyphopodia and abortion of root entry[65,74], which is also observed (although not exclusively) on wild-type roots after ACC or ethephon treatment (Supplementary Fig. 1B). The effect of ethylene on *LYS7*[48] and *CCaMK* is additionally supported by our finding that ethylene signaling seems to act on AM primarily in the root epidermis, where these genes are required for fungal entry (Fig. 2). CCaMK is thought to interpret Myc-Factor-induced Ca²⁺-spiking, and dominant versions of CCaMK are epistatic to the plant perception machinery for AM fungi[58,75]. Importantly, we

could restore full AM colonization in the presence of ACC with ubiquitin promoter-driven, ectopic expression of *CCaMK*, supporting the notion that the decreased expression of CCaMK is at least partially responsible for the ethylene-mediated suppression of AM (Fig. 4B).

Likewise, we were able to rescue colonization of ACC-treated roots by co-application of low concentrations (5 and 10 nM) of the synthetic strigolactone analog GR24^SDS (Fig. 4A), the application of GR24^SDS alone led to only a small increase in root colonization. This suggests that without increased ethylene, the system is saturated with strigolactone, while in the presence of high ethylene levels, strigolactone exudation is limited by reduced strigolactone biosynthetic gene expression similar to observations for rice *phr2* mutants[11]. Furthermore, 5 nM GR24^SDS did not affect SMAX1$_{D2}$ accumulation (Supplementary Fig. 13), confirming that these low concentrations of GR24^SDS mainly affect the fungus. Thus, a reduction in strigolactone biosynthesis and exudation contributes to the ethylene effect on AM. In addition to AM, root nodule symbiosis between legumes and nitrogen-fixing rhizobia employs Common Sym signaling (therefore the name) for signal transduction after perception of rhizobial Nod factors and bacterial entry into the root (summarized in ref. 76). Interestingly, rhizobial Nod factors can induce strigolactone biosynthesis genes in *Medicago truncatula* in a *DMI3/CCaMK*-dependent manner[77]. It is possible that also fungal chitin-oligomer-induced strigolactone biosynthesis gene activation[78] requires CCaMK. Thus, the rescue of colonization of ACC-treated roots by ectopically expressed *CCaMK* may include improved AM fungus-responsive strigolactone biosynthesis.

ACC treatment suppressed AM-marker genes in rice that normally transcriptionally respond to exudates from germinating AM fungal spores[79], in line with our observation that ethylene inhibits expression of the Myc factor perception machinery (*Lys7* and Common Sym genes acting downstream of perception), which is required for induction of these marker genes[79,80]. Also rhizobial Nod Factors are perceived by LysM receptor-like kinases (in *L. japonicus* Nod Factor Receptor (NFR) 1 and 5[50]) inducing Common Sym signaling including Ca$^{2+}$-spiking, activation of CCaMK, and phosphorylation of CYCLOPS (summarized in ref. 76). It was observed that ACC treatment reduces Ca$^{2+}$-spiking in response to Nod factors[81]. In our RNA-seq experiment, expression of *NFR5* and of the ATPase gene *MCA8*, which is required for the generation of Ca$^{2+}$-spiking[82] were reduced in ethephon-treated non-colonized roots (Fig. 2C). This could potentially explain the previously observed reduction in Ca$^{2+}$-spiking in response to Nod factor. However, the observed reduction in Ca$^{2+}$-spiking[81] occurred within about 25 minutes, suggesting that additional posttranslational processes may be influenced by ethylene and may play a role in rapid suppression of Ca$^{2+}$-spiking.

Mutants in the karrikin signaling target SMAX1 boost root colonization by AM fungi (Fig. 6A), although they produce increased amounts of ethylene[21]. Our data showing that root colonization of *smax1* is resistant to ACC treatment (Fig. 6A) and that *ein2a ein2b smax1* triple mutants show the same level of root length colonization as *smax1* and *ein2a ein2b* double mutants provide strong support that *SMAX1* is required for the effect of ethylene on AM. We show that SMAX1 accumulates upon ACC treatment in an *EIN2*-dependent manner (Fig. 6E, F) and targets many of the same genes as ethylene (Fig. 5B–D). Together these data indicate that SMAX1 accumulation and suppression of its direct or indirect targets (strigolactone biosynthesis and common symbiosis genes) is responsible for the effect of ethylene on AM (however, we cannot exclude that ethylene may also alter SMAX1 activity). This interpretation is also consistent with the increased level of transcript accumulation of the karrikin marker gene *DLK2* in *ein2a ein2b* double mutants, resembling *smax1* mutants, and the rescue of the ethylene effect by addition of karrikin, which is known to stimulate SMAX1 degradation[67] (Fig. 6C, F). Ethylene-mediated accumulation of SMAX1 is independent of KAI2 and can be counteracted in a KAI2-dependent manner by co-treatment with

karrikin (Supplementary Fig. 12), indicating that KAI2-mediated degradation can compete with the ethylene-mediated accumulation in the presence of enhanced amounts of KAI2-ligand (KL).

SMAX1 accumulation occurs not only in tissue susceptible to AM fungi (host plant roots) but also in leaves of *N. benthamiana* and roots of the non-host plant *A. thaliana* (Supplementary Fig. 11). This suggests that this phenomenon is widespread in plants and may play a role in a number of ethylene-mediated plant responses, which remain to be discovered. However, ethylene also acts independently of SMAX1 such as in root hair elongation or triple response[21,22].

How might ethylene promote SMAX1 accumulation? One possibility is that ethylene suppresses endogenous KL biosynthesis or promotes KL degradation, thereby reducing activation of the KAI2-MAX2 complex and ubiquitylation of SMAX1. Another reason for SMAX1 accumulation could be the reduced expression of *KAI2* genes (Fig. 3C, Supplementary Fig. 5) upon ethylene treatment. Alternatively, ethylene may increase SMAX1 levels independently of a potential reduction of KAI2-MAX2 activity, e.g., by activating the production of proteins that stabilize SMAX1 through protein-protein interactions or posttranslational modification or by increasing *SMAX1* translation. The mechanism by which ethylene promotes SMAX1 accumulation via EIN2 awaits future discovery.

Based on the transcriptomes of *smax1* mutants and ethephon-treated plants, it appears that a large number of genes with decreased expression upon ethephon treatment do not overlap with genes with increased expression in *smax1* mutants (Fig. 5B, C). If ethylene reduces gene expression by promoting the accumulation of SMAX1, a larger overlap may be expected at first sight. However, when taking together the transcriptome data with SMAX1$_{D2}$-GFP accumulation in nuclei of *L. japonicus* roots (Fig. 6E, F, Supplementary Fig. 12), it is apparent that SMAX1 does not accumulate at high levels and not in all nuclei of wild-type roots in the absence of ethylene. This suggests that not all potential target genes may be fully suppressed by SMAX1 at any given time, which may explain the lower number of genes with increased expression in *smax1* vs. wild type as compared to the number of genes suppressed by ethylene. Instead, a stronger SMAX1 accumulation, like upon ethephon treatment, seems necessary to make the repression of more genes visible in whole-root RNA extracts as compared to non-treated wild-type roots. Additionally, it is possible that SMAX1 acts in concert with alternating co-repressors, the presence of which may determine whether SMAX1 can suppress certain genes or not. It is tempting to speculate that some of these hypothetical co-repressors may also be affected by ethylene. It is also possible that the loss of SMAX1 across multiple root tissues has different effects on gene expression than the epidermis-specific regulation of AM symbiosis by ethylene, which may lead to SMAX1 accumulation in a highly spatially controlled, cell-specific manner. In any case, the regulation of strigolactone biosynthesis genes (*CCD7*, *CCD8*), *LYS7* and Common Sym genes (*CCaMK*, *CYCLOPS*, *DELLA*) via SMAX1 appears to be sufficient to explain the effect of ethylene on AM development (Fig. 4).

Based on our findings, we propose a model by which ethylene and SMAX1 regulate AM symbiosis. Activation of KAI2 promotes proteasomal degradation of SMAX1, increasing strigolactone biosynthesis and common symbiosis signaling through upregulation of the respective genes, thereby promoting AM development. SMAX1 degradation also leads to an increase in ethylene levels[21] ensuring homeostasis in AM development through cycles of SMAX1 stabilization and degradation (Fig. 7A). When ethylene biosynthesis is increased as a result of parallel ethylene biosynthesis-inducing pathways, as may be the case in the presence of certain stressors, the balance is shifted towards SMAX1 accumulation and suppression of AM development (Fig. 7A, B). In summary, we establish an unforeseen molecular link between ethylene and karrikin/KL signaling and provide a molecular explanation for the suppressive effect of ethylene on AM formation.

## Methods

### Plant material and growth conditions

*Lotus japonicus* ecotype Gifu B-129 was used as wild type and as background for all *L. japonicus* mutants. The *ein2a-2 ein2b-1* (cv. Gifu) double mutant was kindly provided by Dr. Dugald Reid and Prof. Jens Stougaard at Aarhus University[34]. Seeds for *ccamk-13* mutants were kindly provided by Prof. Martin Parniske at LMU Munich[83,84] and *smax1-2, smax1-3* and *kai2a kai2b* mutants are from[21,52]. Seeds were scarified and surface-sterilized for 5 min in a sterilization solution (10% Clorox + 0.1% SDS). Sterilized seeds were germinated on a Petri dish containing water agar in dark for 2–3 days at 24 °C in a Sanyo plant growth cabinet (Sanyo, USA). Subsequently, germinated seedlings were exposed to light and grown for 2 weeks in a long-day photoperiod (16 h light/8 h dark) condition at 24 °C. Plants were then transferred to pots filled with quartz sand (described in ref. [85]) or tip-rockwool hydroponics (described in ref. [36]) and supplied with 500 spores per plant of *Rhizophagus irregularis* DAOM 197198 (type C, Agronutrition, Toulouse, France). For experiments with rice, seeds of *Oryza sativa* ssp *japonica* cv. Nipponbare were sterilized by incubating with absolute ethanol for 5 min, incubating in sterilization solution for 30 min and then washing with ddH$_2$O for 30 min. Seeds were germinated for 5 days and rice seedlings were then transferred to hydroponic set-up (described in ref. [36]) in co-cultivation with 500 spores of *Rhizophagus irregularis* DAOM 197198 per plant. All plants co-cultivated with *R. irregularis* were fertilized with a half-Hoagland solution containing 25 μM phosphate in both sand pot and hydroponics set-ups.

The *Arabidopsis thaliana* Col-0 transgenic lines harboring p*UBQ10:SMAX1_{D2}-LUC* (pRATIO2251) or p*SMAX1:SMAX1_{D2}-GFP* (pCAMBIA2300) have been described previously[67,69]. *Arabidopsis* seeds were germinated on half-strength Murashige-Skoog medium (0.5xMS) supplemented with MES buffer and vitamins (Research Products International, Catalog Number M70800-50.0), with 0.8% (w/v) Bacto agar, pH 5.7. Before sowing, seeds were sterilized with 70% (v/v) ethanol containing 0.05% (v/v) Triton X-100 (VWR, Cat. No. 0694-1 L) for 5 minutes with gentle agitation, followed by rinsing with 70% (v/v) and 95% (v/v) ethanol, and air drying.

### Treatment with ethylene, GR24$^{SDS}$ and/or karrikin

For ethylene biosynthesis inhibition or ethylene treatment of *Lotus japonicus* by 2-aminoethoxyvinyl glycine (AVG, Sigma-Aldrich, USA, Cat. No. 32999), 1-aminocyclopropane-1-carboxylic acid (ACC, Sigma-Aldrich, USA, Cat. No. 149101-M) or ethephon (Sigma-Aldrich, USA, Cat. No. C0143) for plants grown in pots, the pots were placed in closed, transparent plastic containers to reduce escape of the gaseous ethylene. ACC, ethephon or AVG were added to the watering solutions (or growth media on Petri dishes) at concentrations indicated in the figures and provided by watering plants with 30 ml of these solutions twice a week for the growth period indicated in figure legends. Stock solutions for all three chemicals were produced with autoclaved ddH$_2$O, which was used as a solvent control for all three chemicals.

For co-treatment with GR24$^{SDS}$ or karrikin, plants were treated (or co-treated with ACC) with GR24$^{SDS}$ (StrigoLab, Italy) or karrikin (OlChemIm, Czech Republic, Cat. No. 0257393) at concentrations indicated in the respective figures. GR24$^{SDS}$ was dissolved at 10 mM concentration in 100% acetone and karrikin at 10 mM concentration in 75% methanol to produce stock solutions and directly added to the Hoagland solution. Control Hoagland solution was supplied with equal amounts of acetone or 75% methanol respectively. Plants were watered 2 times per week with hormone-containing Hoagland solution for 4 weeks. At 4 weeks, roots were harvested for acid-ink staining and quantification of colonization.

### Quantification of root colonization and imaging

Roots were harvested into 10% KOH and heated for 15 minutes at 95 °C followed by incubation in 10% acetic acid, ink staining solution and 5% acetic acid de-staining solution as described[85]. Root pieces of 1 cm each were then mounted on microscopy slides for observation and quantification of colonization at 10X magnification under a light microscope (Leica, type 020-518.500 DM/LS; Leica, Germany). Quantification of AM colonization was performed using a modified gridline intersect method[85,86]. Images of ink-stained roots were taken with a Leica DM6 B microscope (Leica, Germany) equipped with a Leica DFC9000 GT camera.

### RNA extraction and qRT-PCR

Plant roots were harvested and frozen in liquid nitrogen for transfer to -80 °C. Frozen roots were ground to a fine powder in liquid nitrogen, immediately transferred to lysis extraction buffer and processed for RNA isolation using Spectrum™ Plant Total RNA Kit (Sigma-Aldrich, USA). Isolated RNA was treated with in-vitro amplification DNAse (Sigma-Aldrich, USA, Cat. No. AMPD1) and verified for genomic DNA contamination by PCR using genomic-DNA specific primers. RNA samples were quantified with Nanodrop (Thermo Scientific, USA). First-strand cDNA synthesis was carried out with Invitrogen SuperScript III (Thermo Scientific, USA, Cat No. 18080093). qPCR was conducted with primers specified in Supplementary Table 4 and mi-real-time EvaGreen® Master Mix (Metabion, Martinsried, Germany) on a QuantStudio 5 Real-Time PCR System (Applied Biosystems, USA) for Fig. 3C and on a IQ5 Multicolor Real-Time PCR Detection System (BioRad, USA) for Fig S8.

### RNA-sequencing and data analysis

For RNA-seq after ethephon treatment, wild-type and *ein2a ein2b* plants were mock-inoculated or inoculated with 500 spores of *R. irregularis*, grown in hydroponics for 4 weeks and treated with 20 μM ethephon or solvent control. The hydroponics solutions were exchanged twice a month. mRNA was extracted from shock-frozen *L. japonicus* roots as described above. For RNA-seq of *smax1* mutants, wild-type, *smax1-2* and *smax1-3* were grown in pots filled with quartz sand for 5 weeks and fertilized with half Hoagland solution containing 25 μM phosphate, 30 ml once a week. Isolated RNA was treated with DNase from TURBO *DNA-free*™ Kit (Thermo Scientific, USA, Cat No. AM1907) and verified for genomic DNA contamination by PCR using genomic-DNA specific primers. RNA samples were quantified with Nanodrop (Thermo Scientific, USA) followed by quality check with a Bioanalyzer (Agilent, USA) and quantification in Qubit 3.0 fluorometer (Invitrogen, USA). Samples with a RIN value of > 7 were used for library preparation using a QuantSeq 3' library kit (Lexogen, USA) and sequenced on Illumina HiSeq 2500 to obtain 100 bp single end reads. Raw fastq files were processed through trimming (BBDuk, https://jgi.doe.gov/data-and-tools/bbtools/bb-tools-user-guide/bbduk-guide/) to remove the polyA tail from the right end and 12 nucleotides from the left end of each read. Trimmed reads were processed through SubRead package in Conda (mapped onto Lotus v3.0 MG20 genome) and read count quantification via FeatureCounts[87,88]. Raw read counts were subjected to data exploratory (PCA) analysis to conduct differential expression analysis in R software environment (https://www.r-project.org/) using Bioconductor EdgeR package[89]. Differentially expressed genes (DEGs) were obtained with the cut-off: absolute(log$_2$FoldChange) ≥ 0.59 and adjusted p-value ≤ 0.01. Further downstream analyzes consisted of gene-ontology enrichment test (AgriGO v2[90]) and DEGs heatmap.

### Plasmid construction

Genes were amplified with Phusion® High-Fidelity DNA Polymerase (NEB, USA, Cat No. F530S) from cDNA of wild-type *L. japonicus* according to standard protocols and using cloning primers indicated in Supplementary Table 4. Plasmids were constructed using Golden Gate cloning as described before[91] and as indicated in Supplementary Table 5.

## Hairy root transformation

Transgenic hairy roots were induced in *L. japonicus* hypocotyls infiltrated and transformed with LIII plasmids (Supplementary Table 4) using transgenic *A. rhizogenes* AR1193 as described[40]. Transformed roots were screened under a stereo microscope (Leica MZ16 FA; (Leica, Germany) to select roots with mCherry fluorescence.

## SMAX1$_{D2}$ accumulation assay

For SMAX1$_{D2}$ accumulation assay in *Lotus japonicus*, transgenic hairy roots expressing p*Ubi:LjSMAX1$_{D2}$-GFP*_p*35s:mCherry* were generated in wild type, *ein2a ein2b* or *kai2a kai2b* double mutant background. The induced hypocotyls were transferred onto square Petri dishes containing Gamborg B5 medium (Duchefa Biochemie, Netherlands, Cat No. G0209) solidified with 0.8% plant agar and supplemented with 300 µg ml⁻¹ Cefotaxime (TCI, Japan, Cat No. C2224). Petri dishes were partially covered with black paper to keep the roots in the dark and placed vertically at 24 °C with a 16 h light/8 h dark cycle for 3 weeks. Positive transformants, selected with mCherry fluorescence, were transferred to fresh plates with the same medium for 24 hours as pretreatment and then transferred to Petri dishes containing the same medium supplemented with 1 µM ACC or water control; or 3 µM karrikin$_1$, 1 µM ACC and 3 µM karrikin$_1$ or solvent control (0.0075% methanol) for a 24-hour treatment. Roots were excised and their epidermis imaged at the maturation zone 4-6 mm away from the root tip using Leica Stellaris 8 Falcon confocal laser scanning microscope. Sequential scanning for GFP (excitation: 489 nm, detection: 492–530 nm) and mCherry fluorescence (excitation: 587 nm, detection: 590–650 nm) was carried out simultaneously with bright field image acquisition.

Segmentation of cell nuclei in the images and subsequent fluorescence intensity measurements were performed using Imaris 10.0.1 software (Oxford Instruments, UK) in a semi-automated manner. Objects (nuclei) that passed a size-based quality threshold and were in focus were selected for intensity measurements. Image post-processing was done using Fiji (http://fiji.sc/).

For confocal microscopy of *A. thaliana* roots, sterilized seeds of the p*SMAX1:AtSMAX1$_{D2}$-GFP* transgenic lines and Col-0 wild type were plated on 0.5xMS square Petri Dishes (100 mm×100 mm). The seeds were incubated at 4 °C in the dark for three nights, then grown vertically at 21 °C under white light (MaxLite LED T8 4000 K, -110 µmol m⁻² s⁻¹) with long-day photoperiod (16-h light / 8-h dark). After 5 days of growth, some of the seedlings were used for microscopy without treatment, while the remaining seedlings were sprayed with 10 µM ACC or water and incubated for an additional 14 hours prior to microscopy. We performed confocal microscopy as described previously[67]. Briefly, we took images using an inverted confocal microscope (SP5, Leica). Just prior to imaging, seedlings were soaked in 0.1 mg/mL of propidium iodide (Thermo Fisher Scientific, Cat. No. 440300250) in the dark for 3 minutes. GFP fluorescence was excited with a 488 nm laser and detected emission spectra were observed at 510–540 nm (HyD 2 with a voltage of 100 V). Propidium iodide fluorescence was imaged using an excitation wavelength of 543 nm laser with detected emission spectra at 587–625 nm (HyD 4 with a voltage of 100 V).

## Western blot

For *Lotus japonicus*, wildtype hairy roots were transformed with p*Ubi:LjSMAX1$_{D2}$-GFP*_p*35s:mCherry* as described for the microscopic SMAX1$_{D2}$ accumulation assay. Positively transformed roots grown for 5 weeks on selection media were transferred to fresh plates with the same medium for 24 hours as pre-treatment and then transferred to Petri dishes containing the same medium supplemented with 1 µM ACC or water control for a 24-hour treatment. Roots were excised and flash-frozen in liquid nitrogen. The tissue was ground into a fine powder of which 300 mg were used for protein extraction.

For *Nicotiana benthamiana*, leaves were transiently transformed by infiltration with *Agrobacterium tumefaciens* strain AGL1 as described in ref. [65] to express p*Ubi:LjSMAX1-GFP*_p*35s:mCherry*, either without or with co-infiltration of *A. tumefaciens* containing a P19 RNA silencing suppressor expression cassette. After 2 days of incubation, leaves were infiltrated with 1 µM ACC or water control and incubated for 24 hours. Three leaf disks of 1 cm diameter were harvested and flash-frozen in liquid nitrogen. The tissue was ground into a fine powder for protein extraction.

The tissue powder was homogenized in 300 µl lysis buffer (62.6 mM Tris, 2% SDS, 10% Glycerol, 1 mM DTT, 10 µM Bortezomib (StressMarq Biosciences, Canada, Cat. No. SIH-328)) by vortexing. Samples were heated at 95 °C for 5 minutes and then centrifuged at 14,000 g for 30 mins. The resulting supernatant was subjected to SDS page for Western blotting.

Proteins were separated on 10% (w/v) SDS gel using the Biorad Mini-PROTEAN Tetra Vertical Electrophoresis Cell and transferred to 0.45 µm Immobilon®FL PVDF membrane (Millipore) using the Hoefer TE22 Mighty Small Transfer Tank system. The membrane was blocked in 5% skim milk powder (w/v) in PBS (pH 7.4) and probed for 90 minutes with the primary antibody rabbit anti-GFP to detect SMAX1 and SMAX1$_{D2}$ (ChromoTek, Martinsried, Germany, dilution: 1:5000) and rabbit anti-mCherry to detect mCherry (Thermo Fisher Scientific, USA, dilution: 1:5000) in 1% skim milk powder (w/v) dissolved in PBST (PBS, 0.2% Tween-20) at room temperature. The membrane was washed with PBST and incubated with fluorescently labeled goat anti-rabbit secondary antibody (LI-COR Biosciences, Lincoln, USA, dilution: 1:15000) in 1% skim milk powder (w/v) dissolved in PBST for 80 mins at room temperature. After washing, the membrane was imaged by a LICOR Odyssey M imaging system. Image processing and measurement of fluorescence intensities of the protein bands was done using the LICOR Image Studio Lite version 5.2.

## Luciferase reporter assay

Luciferase reporter assays were performed in 96-well plates (Perkin Elmer OptiPlate96). The plates were sterilized by filling each well of the 96-well plate, the corners, and the dome of the plate with 70% (v/v) ethanol, followed by 5-minute incubation, then rinsing with 95% (v/v) ethanol and air drying. For the experiment, each well was filled with 250 µL of 0.5xMS. Using a 200 µL pipette, we placed a single sterilized seed of the p*UBQ10:SMAX1$_{D2}$-LUC* transgenic line in the center of each well. The plates were covered with the sterilized dome, fixed with wide micropore tape, and sealed with two rounds of wide micropore tape. The plated seeds were incubated at 4 °C in the dark for three nights and then grown horizontally at 21 °C under white light (MaxLite LED T8 4000 K, -110 µmol m⁻² s⁻¹) with long-day photoperiod (16-h light / 8-h dark). After 7 days of growth, 20 µL of sterile water was added to each well to prevent drying of the seedlings. The re-sealed plates were incubated for 2 more days in the conditions described above. We then performed the luciferase assays using a modified protocol based on previously described methods[69]. For the luciferase assay, the plate was uncovered and sprayed with 2 mM D-luciferin (GoldBio, Cat. No. LUCK-1G) solution, followed by incubation in the dark at room temperature for 3 hours for equilibration. Each plant was then treated with 50 µL of a solution containing 2 mM D-luciferin and ACC (Cayman Chemical, Cat. No. 16132). The luciferase signal was measured every hour using CLARIOstar plate reader (BMG Labtech) with the following settings: Well scan, spiral avg., diameter 6; Filter setting, 580-80, Gain 3600; Gain adjustment. The relative luminescence intensity of each plant at each time point was calculated relative to time zero.

## Statistics, graphics and reproducibility

Data were statistically analyzed and presented graphically using R statistical environment version 4.1.3 (https://cran.r-project.org/) or

Prism 8.0 (GraphPad, USA), respectively. No statistical method was used to predetermine sample size, however we used sample sizes standard in the field for the experimental approaches used in this study. Except one outlier in the *smax1-3* RNAse data set (Fig. S9B) no data were excluded from the analyzes. The experiments were not randomized. The Investigators were not blinded to allocation during experiments and outcome assessment. All statistical information not included in figure legends can be found in the source data file.

### Reporting summary
Further information on research design is available in the Nature Portfolio Reporting Summary linked to this article.

## Data availability
The RNAseq data generated in this study have been deposited in the SRA database under BioProject accession code PRJNA1133936 (ethephon treatment) and PRJNA1133253 (*smax1*). All other quantitative data generated in this study are published as source data along with this article. Source data are provided with this paper.

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

## Acknowledgements

We are grateful to Christine Wurmser (NGS@TUM) for the preparation of RNA-sequencing libraries and to Abul Khayer for providing promoter sequences of the *L. japonicus* genome for a preliminary computational analysis. We thank Elaine Yeung, Utrecht University, for rice Nipponbare seeds, Dugald Reid and Jens Stougaard for *L. japonicus ein2a-2 ein2b-1* seeds and Franziska Brückner for excellent technical assistance. The study was supported initially by a grant of Valent BioSciences LLC, USA, then by the Emmy Noether Program of the Deutsche Forschungsgemeinschaft (Grant GU1423/1-2) and then by project A02 of the Transregio Collaborative Research Center 356 'Genetic diversity shaping biotic interactions of plants' (491090170), and a core grant from the Max Planck Society to C.G., a DAAD (German Academic Exchange Service) Doctoral Student Fellowship 57381412 to K.V., Japan Society for the Promotion of Science (JSPS) KAKENHI Grants-in-Aid for JSPS Fellows (22KJ3127) to S.O, and US National Science Foundation grant IOS-1856741 to DCN. The funders had no influence on the study design.

## Author contributions

D.D., K.V., and C.G. conceptualized the study. D.D., K.V., S.O., D.C.N., and C.G. designed experiments. D.D., K.V., S.O., and S.T. performed the experiments and analyzed the data with inputs from D.C.N. and C.G. D.D. analyzed the RNAseq data and produced supplementary data files. R.H. performed *L. japonicus* crosses and genotyping. D.D., K.V., and S.O. produced figures, with D.D. assembling most of the final figures with additions from K.V. D.D., K.V., D.C.N., and C.G. wrote the manuscript with input from S.O. K.V., S.O., D.C.N., and C.G. acquired funding.

## Funding

## Competing interests

The authors declare no competing interests.
