## [Transparent Peer Review file · Nature Communications]

Ethylene promotes SMAX1 accumulation to inhibit arbuscular mycorrhiza symbiosis

Corresponding Author: Professor Caroline Gutjahr

Version 0:

Reviewer comments:

Reviewer #1

(Remarks to the Author)

The work of Das et al investigates the link between ethylene and strigolactones signaling by SMAX1 and their role on AM association in *Lotus japonicus*. The first parts of the paper are more descriptive with a lot of gene-expression profiling. I feel that a lot of findings are confirmatory of what has been shown before (in other species), for example the role of ethylene in repressing AM formation and the role of strigolactones on stimulating AM formation by means of degrading SMAX1 protein levels. The novel and exciting part of this work comes at the end, where the authors found that ACC can stabilize SMAX1 levels and thus present a mode of action how ethylene signaling reduces AM formation by stabilizing SMAX1. However, no follow-up experiments on elucidating the mechanism how SMAX1 is stabilized by ethylene, besides it being EIN2-dependent, is presented. I feel the work could have dug a little deeper to elucidate this mechanism.

A possible problem with some of the experiments is the use of a very high level of ACC. Although the authors acknowledge the recent insights that ACC may act as a signal on its own, independent from ethylene, their ACC doses used are all very high. I appreciate the Ethephon experiments, as a crucial confirmation that the observed phenotypes link with ethylene (and not ACC). Nonetheless, some critical experiments to elucidate the crosstalk between ethylene and strigolactones (SMAX1) have been executed with very high ACC levels, probably masking true physiological responses. For example, for the rescue assays in Figure 3, the authors used 200 μM ACC, which is extremely high (2 μM typically is saturating for roots with respect to ethylene production levels from ACC). They also showed that 20 μM was functional (Supl Fig S2). So I wonder, what is the lowest level of ACC (maybe a physiological relevant level of ACC below 2 μM) that suppresses AM formation? I also wonder if GR24 and CCamK can also rescue lower levels of ACC? Furthermore, their ectopic work in *Arabidopsis* show that ACC only stabilizes SMAX1 at low levels (up to 10 μM ACC; Supl Fig S11), and thus they cannot exclude any ACC-specific effects independent of ethylene in other experiments.

I also think that the proposed additive effect of SMAX1 and ethylene (by ACC or EIN2) can be further explored, for example by using either strigolactones inhibitors or ethylene signaling inhibitors (e.g. 1-MCP). This way, the true nature of the ethylene-strigolactones crosstalk can be revealed. It would be relevant to show that a 1-MCP treatment reduces the AM colonization in *smax1* mutants (or not). The other way around, it would be relevant to show that inhibiting strigolactones production (or signaling) reduces the *ein2*-dependent enhanced AM formation.

Some minor comments:

Which gene is meant with the EFE (ethylene forming enzyme) mentioned in line 309. Do the authors mean ACC-oxidase? (to my knowledge, plants do not have a bacterial EFE).

The finding that ACC (not ethylene) stabilizes SAMX1 in an EIN2-dependent way, makes me wonder if SMAX1 is a direct target of EIN2 translational regulation, or that downstream ethylene response can evoke SMAX1 stability? This could be tested by checking SMAX1 protein accumulation in the presence of 1-MCP and ACC.

In line 478, the alternative hypothesis that GR24 can also operate directly by degrading SMAX1, can be easily tested in the ectopic SMAX1-GFP system presented in Figure 5. Is a GR24 treatment degrading SMAX1 proteins in the presence of ACC?

Reviewer #2

(Remarks to the Author)

The findings described in this ms indicate that ethylene suppresses the accumulation of transcripts required for root

colonization by arbuscular mycorrhizal (AM) fungi, including those of strigolactone biosynthesis genes and common symbiosis genes, and inhibits AM symbiosis formation through SMAX1. Furthermore, the results suggest that increased ethylene promotes the accumulation of SMAX1 protein, indicating that SMAX1 functions as a hub that integrates multiple signals in the regulation of AM symbiosis. While this research is highly intriguing and offers valuable insights, I think it is necessary to conduct analysis at the protein level. A more comprehensive understanding could be achieved by delving deeper into the following aspects.

Ethylene signaling pathway analysis
Mutant analysis of ethylene receptors: Analysis of mutants lacking functional ethylene receptors will provide insights into the molecular mechanisms underlying ethylene-mediated inhibition of AM symbiosis.

SMAX1 function analysis

Protein-protein interaction analysis of SMAX1: By analyzing which proteins interact with SMAX1, you can elucidate the downstream signaling pathway of SMAX1.

Subcellular localization analysis of SMAX1: By analyzing the subcellular localization of SMAX1, you can obtain new insights into how SMAX1 functions.

Reviewer #3

(Remarks to the Author)

Revision on Manuscript NCOMMS-24-42745 by Gutjahr and colleagues

This is an interesting manuscript in which the connection between the negative role of ethylene in the arbuscular mycorrhizal symbiosis and the Karrikin signaling has been analyzed. The authors show that the transcriptomic response of ethylene treatment and that of the transcriptional repressor SMAX1 partially overlap in the context of the symbiosis. The authors also show that the detrimental effect of ethylene on the symbiosis requires ethylene perception and signaling. By constitutive expression of one of the key early mycorrhizal genes that are suppressed in response to ethylene or expression of SMAX1, they were able to rescue the repressive effect of ethylene, demonstrating that downregulation of mycorrhizal markers is key in the response to ethylene. Thus, they propose that ethylene signaling goes through SMAX1. Consistent with their hypothesis is the fact that SMAX1 is necessary for the ethylene response. The authors also show some evidence that ethylene signaling increases the amount of SMAX1, concluding that the negative effect of ethylene in the early colonization phase of the symbiosis is mediated by impacting on the accumulation of the SMAX1 repressor. The manuscript is well-written and the results are novel and interesting. However, the main conclusion and title of the paper is in my opinion not sufficiently demonstrated. and more conclusive evidence with additional experiments on how ethylene impacts on SMAX1 would be required.

Specific comments

Line 1, Title. As mentioned above, the title seems to be the main conclusion of this work. However, while the hypothesis that the negative role of ethylene is mediated by SMAX1 is sufficiently clear, that ethylene provokes SMAX1 accumulation should be more thoroughly proven. See below

Line 36, Abstract, I think this abstract could be written in a more comprehensible and attractive manner. In particular, the link between karrikin and strigolactones during the AM symbiosis is not known to all, and as it is written obscures the interplay between the three hormones. As such, the last sentence is a bit devoid of meaning. Rephrasing would help to better convey the important message.

Line 88, In the same line the introduction needs some adjusting. The connection between strigolactones and karrikins during symbiosis should be clearly stated, it is very vague and there is no literature context which is needed to better understand the results shown later.

Line 113, This paragraph comes a bit out of the blue, as there is no obvious link in the previous paragraphs between ethylene and karrikins or their signaling pathways. Perhaps a small hint what led the authors to investigate the possibility that they were linked would help at this point. It should be mentioned the previous findings, in which they show that smax1 mutants have increased levels of ethylene at this point. A suggestion could be to describe first the karrikin and strigolactone roles in AM, the role of SMAX1 as suppressor of symbiosis, and then describe that SMAX1 has increased levels of ethylene. Then, given that it was long ago known that ethylene hinders AM, the authors investigate this apparent paradox.

Line 177, I am not certain what is the meaning of hypermorphic here.

Line 255, The findings summarized in the diagram of Figure 5A and B should be shown in a separate Excel Files, I could not find them. Same for other Venn Diagrams. It would help also the reader to pinpoint the changes that the authors mention.

Line 283, The labeling in the table shown in this Figure is not very comprehensible. Also not clear why black cells do not show any number, it says in the legend because they are not significant, but due the confusing labeling of each column it is not very clear what the meaning of the results is. Even if not significant, the numbers should be shown.

Line 298, I think the authors should elaborate in the meaning of this sentence. Why are those the genes to be looked up? What are the hypotheses they imply?

Line 293 to 299, This is a key finding in this manuscript and as such it should be backed up with experiments that confirm

that indeed the amount or/and activity of SMAX1 is enhanced in response to ethylene, for instance with a quantitative western blot showing increased accumulation of recombinant SMAX1 in the presence of ethylene. In the same line and as alternative, the abundance of SMAX1 protein in ethylene overproducers and ethylene biosynthesis mutants could be shown;

An alternative hypothesis is that ethylene could promote the activity as well as the stability of SMAX1. Could an alternative scenario be that a suppressor of SMAX1 activity is downregulated by ethylene causing activation? The authors suggest the co-repressor possibility later on. Please consider and discuss alternative scenarios. Are the transcriptomic data showing any light in this direction?

Line 512 and 523, This finding is intriguing and as such it cast doubts that the hypothesis is as linear as the authors suggest. Should not the overlap be larger between the two treatments? Or in any case, the amount of genes regulated by ethylene be larger?

Line 515, This is also very puzzling, given that overexpression of DLK2 in tomato has been shown to downregulate key mycorrhizal marker genes (Ho-Plagaro et al.) and thus, DLK2 postulated as a negative regulator of symbiosis. How are the transcriptomic data of this manuscript related to those of Ho-Plagaro? The authors should integrate this into their hypothesis (for example having a look at key orthologue markers) and suggest a possible scenario to explain this apparent contradiction.

Line 542, A very tempting model that should be validated with biochemical data as mentioned above.

Line 544, I think the authors have stretched a bit here regarding the response to stress, given that their results rather suggest a role for this module in controlling the early stages of colonization rather than regulating the progress or maintenance of the symbiosis in the context of stress. Please discuss this point.

Spelling mistakes
Line 101, Arabidopsis

Version 1:

Reviewer comments:

Reviewer #1

(Remarks to the Author)

I appreciate the rebuttal letter and the responses of the authors to my comments. This has clarified most of my questions and concerns. I also applaud the extra work that was done on studying SMAX1 protein levels and the involvement of karakins. The only thing that was passed on to future work is unravelling the molecular mechanism of how EIN2 or ethylene directs SMAX protein stabilization. Although I agree that this might be challenging to address, it would be a key finding to understand how ethylene inhibits AM colonization. The latter would be a major impactful finding, next to the other results presented in the manuscript.

Reviewer #2

(Remarks to the Author)

The study is highly commendable for demonstrating, with extensive data, the connection between ethylene and SMAX1 in inhibiting symbiosis with AM fungi. However, while this paper shows that ethylene suppresses the initiation of AM fungal symbiosis, I wonder whether the state of AM fungi already in symbiotic association with the plant is also influenced by ethylene. If there are any insights on this point, I suggest addressing it in the discussion section.

Reviewer #3

(Remarks to the Author)

This is the revised version of a manuscript I reviewed before. The authors have clarified/corrected most of my points and added new experiments as required that support their original hypothesis of ethylene being a negative regulator of AM symbiosis through the stabilization of the SMAX1 repressor protein. Still a key question remains, also asked by the other referees about the molecular mechanisms of how SMAX1 is stabilized by ethylene, but it is possibly out of the time range for this manuscript. I have a few comments and suggestions that might help to further strength the findings of this interesting manuscript.

1. Given that the defense response hypothesis for the negative impact of ET on AM is rejected, I think this it would be important if these two effects of ET could be more solidly disentangled analyzing the defense data on your transcriptomics. For example, Are smax1 plants ethylene insensitive (to externally added ET) or only the negative effect of ET in AM is missing in these plants? More specifically, are defense responses that happen by ET extra addition observed in smax1 mutants (which have elevated levels of ET), or are they different?

If they are not induced in smax1, why not? Because they need smax1? And then it is possible that this is why AM colonization increases independent of elevated ET levels

If Yes, then indeed defense responses might not be needed for the negative effect of ET on AM. Only the ET induction of PR10 was analyzed in the Suppl. Fig. 8, what about in the smax1? What about all other defense genes induced by ET?

2. The experiment in Nicotiana leaves, as well as the one in Arabidopsis demonstrates that besides its function in the mycorrhizal symbiosis, conserved mechanisms exist that leads to the stabilization of SMAX1 upon ET treatment, not only in plants in general but also accross tissues, therefore likely impacting on the transcription/repression of genes under the control of SMAX1. Thus, I think this is important to highlight this even more in the discussion, because it has far more consequences than just affecting the AM symbiosis.

And again, how this stabilization upon ET perception occurs is the most important question. The authors offer many interesting hypotheses in the discussion, but it would be good also to highlight some if they are supported by their transcriptomic data. For example, are there hints of transcriptional activation of putative KL catabolic enzymes? I think the hypothesis that the authors explained in their rebuttal letter about the putative function of DLK2 inactivating KLs would nicely fit into that discussion paragraph, but possibly there are other interesting candidates that would suit the model.

Similarly, when they talk about putative co-repressors in Line 558 of the discussion, are there some candidates identified in the transcriptomic data?

Responses to reviewer comments

We thank the reviewers for their time spent in careful consideration of the manuscript and their helpful comments.

Ethylene suppression of AM development has been documented repeatedly over the last four decades, but the molecular events underlying this inhibition have not been elucidated. While it has been suggested that ethylene-induced defense responses cause suppression of AM, we have discovered that ethylene instead reduces expression of a set of genes required for arbuscular mycorrhiza development. These ethylene-repressed genes include strigolactone biosynthesis genes, which are required to activate the fungus in the rhizosphere with exuded strigolactone. We demonstrate the relevance of this finding by restoring root colonization by AMF through addition of strigolactone at very low dosages that are sufficient to trigger fungal branching. Ethylene also suppressed the expression of Common Symbiosis genes required for intracellular accommodation of the fungus. A constitutively expressed central player in Common Symbiosis signaling called Calcium and Calmodulin dependent protein Kinase (CCaMK) could overcome the ethylene suppression.

We identified SMAX1 as a key integrator of ethylene and karrikin signaling (this work) that suppresses both strigolactone biosynthesis and Common Symbiosis gene expression (this work and, Choi et al 2020). We discovered that SMAX1 protein accumulates upon ethylene treatment, revealing it as a regulatory hub that integrates ethylene and karrikin signaling. We propose that ethylene promotes SMAX1 accumulation and thereby the suppression of genes required for AM development. Thus, we solve a long-standing enigma of how ethylene suppresses AM symbiosis. and remove the old assumption that defense was responsible for AM suppression by ethylene.

We share the reviewers' interest in the biochemical mechanism by which ethylene causes accumulation of SMAX1. We plan to investigate this question, which will require substantial experimentation, over the next several years.

In response to the reviewers' comments, we have strengthened our prior findings with three new lines of evidence:

- 1) We expressed the gain-of-function EIN2 version EIN2B-CEND, the C-terminus of EIN2B, which activates ethylene downstream responses in the absence of ethylene, in the root epidermis of a *smax1* mutant. In wild type plants, this transgene causes suppression of AM colonization. We show that AM colonization of *smax1* is resistant to EIN2B-CEND (just like AM colonization of *smax1* is resistant to ethylene). This supports that SMAX1 acts downstream of ethylene signaling via EIN2 in the root epidermis.
- 2) We show by Western Blot that SMAX1_{D2} accumulates in *L. japonicus* hairy roots and that full-length SMAX1 accumulates in *Nicotiana benthamiana* leaves after ACC treatment. This supports the stabilization of SMAX1 protein by ethylene or ACC.
- 3) We show that accumulation of SMAX1_{D2} in *L. japonicus* roots upon ACC treatment can be reversed by karrikin treatment and this is dependent on the karrikin receptor KAI2. In addition, KAI2 is not required for the ethylene mediated accumulation of SMAX1. This indicates that KAI2-mediated signaling (e.g. of karrikins or endogenous KAI2

ligand) can regulate SMAX1 protein abundance independently and oppositely to ethylene.

Reviewer #1 (Remarks to the Author):

The work of Das et al investigates the link between ethylene and strigolactones signaling by SMAX1 and their role on AM association in *Lotus japonicus*.

The first parts of the paper are more descriptive with a lot of gene-expression profiling. I feel that a lot of findings are confirmatory of what has been shown before (in other species), for example the role of ethylene in repressing AM formation and the role of strigolactones on stimulating AM formation by means of degrading SMAX1 protein levels.

The novel and exciting part of this work comes at the end, where the authors found that ACC can stabilize SMAX1 levels and thus present a mode of action how ethylene signaling reduces AM formation by stabilizing SMAX1. However, no follow-up experiments on elucidating the mechanism how SMAX1 is stabilized by ethylene, besides it being EIN2-dependent, is presented. I feel the work could have dug a little deeper to elucidate this mechanism.

Reply: We wish to clarify that while SMAX1 regulates the expression of strigolactone biosynthesis genes, which has an impact on AM symbiosis, we are not proposing that strigolactone regulates SMAX1 abundance. Degradation of SMAX1 after strigolactone perception may occur, particularly when exogenous strigolactone analogs are applied at sufficient concentration, but it is more typical that SMAX1 is degraded after perception of karrikins or KAI2 ligand(s) by KAI2 (also known as HTL, or D14L in rice).

We respectfully disagree with the assessment that novelty is limited in the first parts of the manuscript. The transcriptomic data was essential for linking two previously independent concepts: 1) that ethylene regulates AM symbiosis and 2) that SMAX1 regulates AM symbiosis. While each overarching phenomena has been recognized, how these mechanisms may work and how they are related to each other are novel components of this work.

We intend to investigate the mechanism of how SMAX1 is stabilized by ethylene, but that will be a large study in itself to be carried out over the next several years.

A possible problem with some of the experiments is the use of a very high level of ACC. Although the authors acknowledge the recent insights that ACC may act as a signal on its own, independent from ethylene, their ACC doses used are all very high. I appreciate the Ethephon experiments, [as a crucial confirmation that the observed phenotypes link with ethylene (and not ACC). Nonetheless, some critical experiments to elucidate the crosstalk between ethylene and strigolactones (SMAX1) have been executed with very high ACC levels, probably masking true physiological responses. For example, for the rescue assays in Figure 3, the authors used 200 μM ACC, which is extremely high (2 μM typically is saturating for roots with respect to ethylene production levels from ACC). They also showed that 20 μM was functional (Supl Fig S2). So I wonder, what is the lowest level of ACC (maybe a physiological relevant level of ACC below 2 μM) that suppresses AM formation?

Reply: We agree entirely that 2 μM ACC is sufficient to observe ethylene effects on agar plates and with 5-day old *Arabidopsis* seedlings, or when roots are incubated directly in ACC-containing solutions. As noted in the manuscript, higher concentrations of ACC or ethephon are required to see significant effects in mycorrhizal plants grown for several weeks in open pots in sand (see results section and materials and methods). We think that low plant availability of ACC in sand and/or escape of gaseous ethylene from this growth medium reduces the ethylene levels actually sensed by the plant (likely far below 200 μM). In preliminary experiments, lower concentrations of ACC or ethephon were insufficient to suppress AM reproducibly. The 20 μM ethephon concentration (Suppl Fig S2) was sufficient for plants grown in hydroponics, in which the liquid medium is in direct contact with the roots. Please note that we have confirmed that ethylene and not ACC causes the effect by examining the AM phenotype of the *ein2a ein2b* mutant. Colonization of *ein2a ein2b* was insensitive to the treatments. To our knowledge, EIN2 is not involved in ACC signaling (Yin et al 2019, J Exp Bot).

I also wonder if GR24 and CCamK can also rescue lower levels of ACC?

Reply: Because lower levels of ACC do not reliably suppress AM in open pots filled with sand we did not perform this experiment.

Furthermore, their ectopic work in *Arabidopsis* show that ACC only stabilizes SMAX1 at low levels (up to 10 μM ACC; Supl Fig S11), and thus they cannot exclude any ACC-specific effects independent of ethylene in other experiments.

Reply: In these experiments seedlings are directly treated with the liquid ACC solution, such that the roots can easily and immediately take up the liquid including ACC. Imaging occurs in very short time scales (after 24h) compared to AM experiments, in which plants grow for 4-6 weeks. In *Lotus* we even saw SMAX1-D2 accumulation with direct treatment of 1 μM ACC for 24h (see Fig. 5 and S12). The concentrations used for different experimental set-ups conducted in different media and at very different time-scales are thus not comparable.

Root colonization of *Lotus ein2a ein2b* was insensitive to ACC and to ethephon. We also found that SMAX1 accumulation did not occur in *ein2a ein2b* roots in response to ACC treatment. These two observations indicate that there is not an ethylene signaling-independent effect of ACC on root colonization or SMAX1 abundance.

The experiments with *Arabidopsis* were designed to determine whether accumulation of SMAX1 in response to ACC is conserved in a plant that lost the ability to form arbuscular mycorrhiza. The reviewer is correct that we cannot say for certain that the response in *Arabidopsis* is due to ethylene and not ACC itself; however, given the observations in *Lotus*, we anticipate that ACC-stimulated stabilization of SMAX1 occurs in a similar manner (via ethylene).

I also think that the proposed additive effect of SMAX1 and ethylene (by ACC or EIN2) can be further explored, for example by using either strigolactones inhibitors or ethylene signaling

inhibitors (e.g. 1-MCP). This way, the true nature of the ethylene-strigolactones crosstalk can be revealed.

Reply: We currently do not have any evidence for an additive effect of SMAX1 and ethylene as the *ein2a ein2b smax1* triple mutant has the same AM phenotype as the *ein2a ein2b* double and the *smax1* single mutants. We note in the manuscript: "Furthermore, *ein2a ein2b smax1* triple mutants displayed root colonization levels that were similar to *smax1* single and *ein2a ein2b* double mutants, and not additive (Fig. 5B)." We must also emphasize that SMAX1 degradation is not necessarily regulated by strigolactone, but has instead been more commonly associated with karrikin/KAI2 ligand signaling. At the moment, there are no known inhibitors of karrikin/KAI2 ligand metabolism.

It would be relevant to show that a 1-MCP treatment reduces the AM colonization in *smax1* mutants (or not).

Reply: Because ethylene reduces AM colonization and 1-MCP is an inhibitor of ethylene perception, it seems unlikely that 1-MCP would have the same AM-inhibiting effect as ethylene or ACC; rather, the opposite effect may be expected. We also note that the AM colonization phenotype of the *ein2a ein2b smax1* triple mutant is the same as *smax1* and *ein2a ein2b*. The lack of additivity in the triple mutant suggests that SMAX1 and ethylene signaling are acting through the same pathway rather than independent pathways.

The other way around, it would be relevant to show that inhibiting strigolactones production (or signaling) reduces the *ein2*-dependent enhanced AM formation.

Reply: We agree this would be interesting, although more factors than just strigolactone production probably play a role in enhancing AM in the *ein2* mutant background. Unfortunately, strigolactone biosynthesis and receptor mutants in *Lotus japonicus* hardly produce flowers. This makes it extremely difficult for us to perform crosses and meaningful experiments with sufficient replication in a reasonable time frame. The reviewer may be aware that the fungal strigolactone receptor and associated signal transduction in the fungus are yet unknown (plus AM fungi are not yet amenable to genetic manipulation), which prevents us from reducing strigolactone signaling in this system. We have shown that addition of extremely low levels of a strigolactone analog, which are sufficient to stimulate the fungus (Besserer et al 2006) are sufficient to rescue the effect of ACC on colonization.

Some minor comments:

Which gene is meant with the EFE (ethylene forming enzyme) mentioned in line 309. Do the authors mean ACC-oxidase? (to my knowledge, plants do not have a bacterial EFE).

Reply: Thank you for this comment. We have changed the name of the gene.

The finding that ACC (not ethylene) stabilizes SAMX1 in an EIN2-dependent way, makes me

wonder if SMAX1 is a direct target of EIN2 translational regulation, or that downstream ethylene response can evoke SMAX1 stability? This could be tested by checking SMAX1 protein accumulation in the presence of 1-MCP and ACC.

Reply: We agree that it is highly interesting to understand the molecular mechanism of ethylene-mediated accumulation of SMAX1 in the future. However, we do not understand how 1-MCP treatment can help us to dissect between a direct or an indirect involvement of EIN2. To our knowledge EIN2 remains inactive when 1-MCP binds to the ethylene receptor. Therefore, this would simply phenocopy the *ein2a ein2b* mutant.

In line 478, the alternative hypothesis that GR24 can also operate directly by degrading SMAX1, can be easily tested in the ectopic SMAX1-GFP system presented in Figure 5. Is a GR24 treatment degrading SMAX1 proteins in the presence of ACC?

Reply: Thank you for the comment. It made us realize that this suggestion in the discussion is unrealistic. Therefore, we removed it. The concentrations of 5 nM and 10 nM GR24 are sufficient to activate the fungus but far too low to enter the plant from sand in sufficient concentrations and to induce SMAX1 degradation. Usually, we start to see plant transcriptional and developmental responses to GR24 treatments at concentrations of 1-3 μ M.

We added a new experiment, in which we co-treated roots with ACC and karrikin. Karrikin counteracted the effect of ACC on SMAX1 accumulation in a *kai2a kai2b* dependent manner (see Fig. S12).

Reviewer #2 (Remarks to the Author):

The findings described in this ms indicate that ethylene suppresses the accumulation of transcripts required for root colonization by arbuscular mycorrhizal (AM) fungi, including those of strigolactone biosynthesis genes and common symbiosis genes, and inhibits AM symbiosis formation through SMAX1. Furthermore, the results suggest that increased ethylene promotes the accumulation of SMAX1 protein, indicating that SMAX1 functions as a hub that integrates multiple signals in the regulation of AM symbiosis. While this research is highly intriguing and offers valuable insights, I think it is necessary to conduct analysis at the protein level. A more comprehensive understanding could be achieved by delving deeper into the following aspects.

Ethylene signaling pathway analysis
Mutant analysis of ethylene receptors: Analysis of mutants lacking functional ethylene receptors will provide insights into the molecular mechanisms underlying ethylene-mediated inhibition of AM symbiosis.

Reply: We have used the *ein2a ein2b* mutant in our work. EIN2 functions as a key signaling component linking ethylene perception to primary transcription factors like EIN3 and EIL1. The *ein2a ein2b* mutant has a similar effect - ethylene insensitivity - as would be expected from loss-of-function mutations in all of the ethylene receptors. Therefore, we are not certain what

new insights would be gained that would justify the long-term work of generating ethylene receptor mutants in *Lotus japonicus*.

SMAX1 function analysis Protein-protein interaction analysis of SMAX1: By analyzing which proteins interact with SMAX1, you can elucidate the downstream signaling pathway of SMAX1. **Subcellular localization analysis of SMAX1:** By analyzing the subcellular localization of SMAX1, you can obtain new insights into how SMAX1 functions.

Reply: Our work presented here addresses the molecular events underlying the suppressive effect of ethylene on AM. Thus, although identifying interactors of SMAX1 and events downstream of SMAX1 is of high interest for other projects, this was not the goal of this study and is beyond the scope of this work.

The subcellular localization of SMAX1 is known. It localizes to the nucleus (e.g. Carbonnel et al 2020, PNAS; Khosla et al 2020, Plant Cell and a number of other articles in other species). In figures 5E, S11D and S12 of the present manuscript, we also show that SMAX1-D2 is localized to the nucleus.

Reviewer #3 (Remarks to the Author):

Revision on Manuscript NCOMMS-24-42745 by Gutjahr and colleagues

This is an interesting manuscript in which the connection between the negative role of ethylene in the arbuscular mycorrhizal symbiosis and the Karrikin signaling has been analyzed. The authors show that the transcriptomic response of ethylene treatment and that of the transcriptional repressor SMAX1 partially overlap in the context of the symbiosis. The authors also show that the detrimental effect of ethylene on the symbiosis requires ethylene perception and signaling. By constitutive expression of one of the key early mycorrhizal genes that are suppressed in response to ethylene or expression of SMAX1, they were able to rescue the repressive effect of ethylene, demonstrating that downregulation of mycorrhizal markers is key in the response to ethylene.

Reply: We thank the reviewer for the thoughtful and constructive comments on our work.

Thus, they propose that ethylene signaling goes through SMAX1. Consistent with their hypothesis is the fact that SMAX1 is necessary for the ethylene response. The authors also show some evidence that ethylene signaling increases the amount of SMAX1, concluding that the negative effect of ethylene in the early colonization phase of the symbiosis is mediated by impacting on the accumulation of the SMAX1 repressor.

The manuscript is well-written and the results are novel and interesting. However, the main conclusion and title of the paper is in my opinion not sufficiently demonstrated. and more conclusive evidence with additional experiments on how ethylene impacts on SMAX1 would be required.

Reply: We have added additional lines of evidence for the impact of ethylene on SMAX1: 1) We show that expression of EIN2-CEND in the root epidermis reduces colonization of WT but not of *smax1*. 2) We show that SMAX1 accumulation in response to ACC is independent of the karrikin/KL receptor KAI2. Furthermore, it can be reverted by karrikin treatment in a KAI2-dependent manner. 3) We also confirm the ethylene-mediated accumulation of SMAX1-D2 in *Lotus* roots and full-length SMAX1 in *Nicotiana* leaves by Western Blot.

Specific comments

Line 1, Title. As mentioned above, the title seems to be the main conclusion of this work. However, while the hypothesis that the negative role of ethylene is mediated by SMAX1 is sufficiently clear, that ethylene provokes SMAX1 accumulation should be more thoroughly proven. See below

Line 36, Abstract, I think this abstract could be written in a more comprehensible and attractive manner. In particular, the link between karrikin and strigolactones during the AM symbiosis is not known to all, and as it is written obscures the interplay between the three hormones. As such, the last sentence is a bit devoid of meaning. Rephrasing would help to better convey the important message.

Reply: Thank you for pointing this out. We have re-written the abstract to increase clarity.

Line 88, In the same line the introduction needs some adjusting. The connection between strigolactones and karrikins during symbiosis should be clearly stated, it is very vague and there is no literature context which is needed to better understand the results shown later.

Reply: Thank you for pointing this out. We have exchanged the paragraphs on karrikin and ethylene signaling to better clarify the connection of karrikin signaling with the mechanisms regulating AM described in the beginning of the introduction. Thereby we have also clarified better the connection between as well as the relevance of karrikin and strigolactone signaling in AM symbiosis.

Line 113, This paragraph comes a bit out of the blue, as there is no obvious link in the previous paragraphs between ethylene and karrikins or their signaling pathways. Perhaps a small hint what led the authors to investigate the possibility that they were linked would help at this point. It should be mentioned the previous findings, in which they show that *smax1* mutants have increased levels of ethylene at this point. A suggestion could be to describe first the karrikin and strigolactone roles in AM, the role of SMAX1 as suppressor of symbiosis, and then describe that SMAX1 has increased levels of ethylene. Then, given that it was long ago known that ethylene hinders AM, the authors investigate this apparent paradox.

Reply: Thank you for this useful comment. We have re-arranged the introduction as suggested.

Line 177, I am not certain what is the meaning of hypermorphic here.

Reply: The term hypermorphic is one of "Muller's morphs." It is one of a classic set of terms that are used to describe the effects of a mutation on gene product function (e.g. hypomorph, amorph, antimorph, hypermorph, neomorph). An allele is hypermorphic when the gene product has an increased level of activity and/or abundance. We have added an explanation to the manuscript text.

Line 255, The findings summarized in the diagram of Figure 5A and B should be shown in a separate Excel Files, I could not find them. Same for other Venn Diagrams. It would help also the reader to pinpoint the changes that the authors mention.

Reply: Thank you for this suggestion. We have added supplemental data files with the gene lists represented in all Venn diagrams (Supplemental Data 3, 10 and 11).

Line 283, The labeling in the table shown in this Figure is not very comprehensible. Also not clear why black cells do not show any number, it says in the legend because they are not significant, but due the confusing labeling of each column it is not very clear what the meaning of the results is. Even if not significant, the numbers should be shown.

Reply: Thank you for pointing this out. We have simplified the labelling of the columns by moving the genotype labels to the top of the heatmap. We prefer not to show the numbers for non-significant gene expression because we would like to highlight only those expression changes that carry meaning (to avoid confusing the reader). If it seems appropriate for the sake of clarity, we could also remove the numbers altogether. However, we thought it was useful to see the actual fold-changes of genes with statistically significant changing in expression.

Line 298, I think the authors should elaborate in the meaning of this sentence. Why are those the genes to be looked up? What are the hypotheses they imply?

Reply: Thank you for this comment. We added the following text to provide a concise example: "for example to identify novel LysM receptors involved in symbiotic signaling".

Line 293 to 299, This is a key finding in this manuscript and as such it should be backed up with experiments that confirm that indeed the amount or/and activity of SMAX1 is enhanced in response to ethylene, for instance with a quantitative western blot showing increased accumulation of recombinant SMAX1 in the presence of ethylene. In the same line and as alternative, the abundance of SMAX1 protein in ethylene overproducers and ethylene biosynthesis mutants could be shown;

Reply: Thank you for this suggestion. We have now performed semi-quantitative Western Blots based on fluorescence (Fig. S11B). The Western Blots are semi-quantitative because we are not able to measure the absolute amount of protein, but instead we measured the relative amount of a protein compared to a transformation control. We expressed SMAX1_{D2} in *L. japonicus* hairy roots and reported the ratio of SMAX1_(D2)-GFP fluorescence to the mCherry transformation marker.

We did not succeed in performing Western blots for full-length SMAX1 expressed in *L. japonicus* roots because the protein appears to be very unstable (as also reported for *A. thaliana*, Khoshla et al 2020, Li et al 2022). However, we were able to confirm the results with full-length SMAX1 expressed in *N. benthamiana* leaves.

In both cases ACC treatment led to a clear accumulation of SMAX1.

Ethylene overproducing and biosynthesis mutants are currently not available for *Lotus japonicus*. We also expect that in a constantly ethylene deficient or overproducing mutant SMAX1 may be subject to feedback regulation or regulation of alternative signals, which may make the result more difficult to interpret than after time-limited ethylene treatment.

An alternative hypothesis is that ethylene could promote the activity as well as the stability of SMAX1. Could an alternative scenario be that a suppressor of SMAX1 activity is downregulated by ethylene causing activation? The authors suggest the co-repressor possibility later on. Please consider and discuss alternative scenarios. Are the transcriptomic data showing any light in this direction?

Reply: Currently we cannot make any statement about SMAX1 activity because no data on this are available.

The only currently known SMAX1 co-repressors are TOPLESS and TOPLESS-RELATED proteins. There may be more. We did not see any *TOPLESS* gene among the DEGS with reduced expression upon ethephon treatment.

Of course, we agree with the reviewer that in addition to SMAX1 accumulation, the activity of the already accumulated SMAX1 protein could be changed. We have mentioned this alternative hypothesis briefly in the discussion. However, since it is very speculative and not backed up by data, we prefer to limit this discussion to a short sentence.

Line 512 and 523, This finding is intriguing and as such it casts doubts that the hypothesis is as linear as the authors suggest. Should not the overlap be larger between the two treatments? Or in any case, the amount of genes regulated by ethylene be larger?

Reply: We agree that this notion is intriguing. Considering our results from imaging of SMAX1_{D2} accumulation (Fig. 5D and Fig. S12) this makes sense: We see rather little SMAX1_{D2} accumulation in solvent-treated *L. japonicus* roots and hardly any SMAX1 accumulation in *N. benthamiana* leaves. This indicates that under conditions of low ethylene presence, SMAX1 abundance is not very high – potentially because of KL-presence and KAI2-activity in many

cells. Therefore, in the *smax1* mutant only a subset of potential target genes may show increased expression in comparison to WT, because many of them are already de-repressed in the WT (due to low SMAX1 abundance). In the presence of ethylene SMAX1 strongly accumulates. This may enable it to suppress more of its potential target genes than in the non-treated WT, the expression of which is significantly reduced after ethephon treatment in the whole root. Whatever the case, the genetics tell us that SMAX1 is required for the suppression of AM by ethylene, since ACC and the expression of EIN2-CEND do not suppress AM in the *smax1* mutant background. Based on our data we think that (citing from the manuscript): “In any case, the regulation of strigolactone biosynthesis genes (*CCD7*, *CCD8*), *Lys7* and Common Sym genes (*CCaMK*, *CYCLOPS*, *DELLA*) via SMAX1 appears to be sufficient to explain the effect of ethylene on AM development (Fig. 3)”. Although it is of course possible that ethylene also regulates other factors that may be involved in suppressing AM. Further insights including the set of direct SMAX1 targets will be obtained in future projects.

Line 515, This is also very puzzling, given that overexpression of DLK2 in tomato has been shown to downregulate key mycorrhizal marker genes (Ho-Plagaro et al.) and thus, DLK2 postulated as a negative regulator of symbiosis. How are the transcriptomic data of this manuscript related to those of Ho-Plagaro? The authors should integrate this into their hypothesis (for example having a look at key orthologue markers) and suggest a possible scenario to explain this apparent contradiction.

Reply: *DLK2* is induced during AM across several species and also in *Lotus japonicus* (see the wealth of published transcriptome data from AM roots across several species). The promoter seems to be active in arbuscule-containing cells in tomato (Ho Plagaro et al 2020) and rice (Sisaphaithong et al 2021). It is possible that DLK2 acts in a negative feedback mechanism to limit AM symbiosis. If the feedback would affect events upstream of SMAX1. In fact, one favored hypothesis in the karrikin signaling community states that it may hydrolyse the endogenous KAI2 ligand, thereby attenuating the signaling (see e. g. Vegh et al 2017, Front Plant Sci). This would be consistent with *smax1* mutant AM phenotypes (increased colonization) not being affected by an increased expression of *DLK2*. In addition, in our view it will be important to confirm the data of Ho-Plagaro et al (currently based on RNAi and ectopic expression) with a real *dlk2* mutant, ideally in multiple species that engage in AM symbiosis.

Line 542, A very tempting model that should be validated with biochemical data as mentioned above.

Reply: Thank you for this suggestion. We have performed Western blots to validate this as described above.

Line 544, I think the authors have stretched a bit here regarding the response to stress, given that their results rather suggest a role for this module in controlling the early stages of colonization rather than regulating the progress or maintenance of the symbiosis in the context of stress. Please discuss this point.

Reply: We do not (and did not intend to) mention ‘maintenance of the symbiosis’ in this paragraph. Root colonization by AM fungi is not static. It is a cycle of colonization, degradation and re-colonization. Therefore, there are always opportunities to suppress colonization when new colonization units start at the epidermis. Of course, (as for all working models) this is a conceptual model that places our data in a conceptual framework of why this regulatory module may have evolved and how this could work in nature. To avoid any confusion, we have toned down this line by using a more conditional form.

Spelling mistakes
Line 101, Arabidopsis

Reply: Thank you, the spelling mistake has been corrected.